# A NEW LAGRANGIAN BASED SHORT TERM PREDICTION METHODOLOGY FOR HF RADAR CURRENTS

Lohitzune Solabarrieta[1,2], Ismael Hernandez-Carrasco[3], Anna Rubio[2], Michael Campbell[1], Ganix Esnaola[4,5], Julien Mader[2], Burton H. Jones[1], Alejandro Orfila[3]

(1) KAUST, Red Sea Research Center, Integrated Ocean Processes, Saudi Arabia.
(2) AZTI Marine Research, Basque Research and Technology Alliance (BRTA), Pasaia, Spain
(3) Instituto Mediterráneo de Estudios Avanzados. IMEDEA (CSIC-UIB), 07190 Esporles, Spain.
(4) Nuclear Engineering and Fluid Mechanics Dept., UPV ,20018-Donostia, Spain.
(5) Joint Research Unit BEGIK, (IEO)- (UPV/EHU), 48620-Plentzia, Spain.

Corresponding author's email: lsolabarrieta@azti.es

**ABSTRACT**

The use of High Frequency Radar (HFR) data is increasing worldwide for different applications in the field of operational oceanography and data assimilation, as it provides real-time coastal surface currents at high temporal and spatial resolution. In this work, a Lagrangian based empirical real-time, Short-Term Prediction (L-STP) system is presented in order to provide short term forecasts of up to 48 hours of ocean currents. The method is based on finding historical analogues of Lagrangian trajectories obtained from HFR surface currents. Then, assuming that the present state will follow the same temporal evolution as the historical analogue did, we perform the forecast. The method is applied to two HFR systems covering two areas with different dynamical characteristics: the southeast Bay of Biscay and the central Red Sea. A comparison of the L-STP methodology with predictions based on persistence and reference fields are performed in order to quantify the error introduced by this approach. Furthermore, a sensitivity analysis has been addressed to determine the limit of applicability of the methodology regarding the temporal horizon of Lagrangian prediction. A real-time skill-score has been developed using the results of this analysis, which allows to identify periods when the short-term prediction performance is more likely to be low, and persistence can be used as a better predictor for the future currents.

## 1. INTRODUCTION

The coastal zone is under increasing human pressure. During recent decades coastal seas have been experiencing intensified activity for recreation, transport, fisheries and marine-related energy production, which, in many cases, results in serious damage to coastal marine ecosystems. A better understanding of the dynamical processes responsible for the surface oceanic transport is a prerequisite for the efficient management of the coastal ocean. Coastal processes are responsible for the transport and fate of multi-source pollutants like plastics, nutrients, jellyfish, harmful algal blooms, etc. Thus, improving the capacity of monitoring and forecasting the coastal area is key for the integrated assessment of the marine ecosystem. This requirement is driving the set-up of a growing number of multi-platform operational observatories designed for continuous monitoring of the coastal ocean from international or national (e.g., US IOOS, EU EOOS, Australian IMOS, etc.) to local scales. Moreover, due to the need of forecasting applications for response to emergency situations such as oil spills, or search and rescue operations, many of the existing operational observatories are linked with operational ocean forecasting models with or without data assimilation (e.g. MARACOOS, NOAA Global Real-Time Ocean Forecast System, COPERNICUS Marine Environment Monitoring System).

With the need of providing a long-term framework for the development and improvement of the European Marine coastal observations, the JERICO Research Infrastructure (JERICO-RI) has been developing methods and tools (through JERICO, JERICO-NEXT and JERICO-S3 projects) for the production of high-quality marine data, and sharing expertise and infrastructures between the existing observatories in Europe. Typically constituted with different in-situ point-wise observational platforms (such as moored buoys, tidal gauges, drifting buoys, etc.) a significant number of these observatories now employ land-based High Frequency Radars (HFR) that provide real-time coastal currents with unprecedented coverage and resolution (e.g. Paduan and Rosenfeld, 1996; Kohut and Glenn, 2003; Abascal et al., 2009; Solabarrieta et al., 2014, Rubio et al. 2017; Paduan and Washburn, 2013). Each HFR coastal site measures radial surface currents moving away or approaching the antenna, based on the shift of the first peak (Bragg peak) of the Doppler spectra (Crombie 1955, Barrick et al 1977). Combining the overlapping radial vectors from at least 2 antennas provides surface true vector currents (Barrick et al., 1977, Barrick and Lipa, 1979). Several studies have compared *in-situ* current measurements with HFR observations (e.g., Schott et al. 1985; Hammond et al. 1987; Paduan and Rosenfeld 1996, Emery et al. 2004; Paduan et al., 2006; Ohlmann et al. 2007; Liu et al., 2014; Solabarrieta et al, 2014, Bellomo

et al., 2015; Lana et al., 2016; Hernandez-Carrasco et al., 2018b) and have
repeatedly demonstrated the potential of this technology. Presently, more than 250
HFR antennas are installed and active worldwide (Roarty et al., 2019; http://global-
hfradar.org/).
Due to their high spatio-temporal resolution, HFR data are commonly used in real
time for search and rescue (Ullman et al., 2006) or oil spill prediction/mitigation
emergency response (Abascal et al., 2017). In addition, there have been several
efforts dedicated to the development of assimilation strategies that incorporate the
HFR measured surface currents into ocean coastal models (Breivik and Saetra,
2001, Oke et al 2002, Paduan and Shulman 2004, Stanev et al., 2011, Barth et al.,
2011), some of which have been tested for short periods of time (Chao et al.,
2009). However, assimilation of HFR data into models is still a computationally
expensive and complex issue, not to mention operational capabilities of such a
procedure. Because of these constraints, the availability of real-time high-
resolution HFR current fields has led to alternative solutions in order to obtain
short term prediction (STP) of surface coastal currents, through the direct use of
HFR historical and nowcast observations using different approaches (e.g. Zelenke
2005, Frolov et al. 2012, Barrick et al., 2012, Orfila et al. 2015, Solabarrieta et al.
2016, Vilibić et al, 2016, Ren et al., 2019, see Table 1).
The above-mentioned studies develop and implement different STP approaches
(harmonic analysis of the last hours, genetic algorithms, numerical models, …)
which often require either additional data, or long training periods of data without
gaps. Hardware failures due to power issues, communications or environmental
conditions often result in spatio-temporal gaps within HFR datasets. Spatial gaps
can be filled on a real-time basis but filling long temporal gaps is not
straightforward. Several gap-filling methodologies have been developed for HFR
data sets: Open Modal Analysis, (OMA) (Kaplan and Lekien, 2007), Data
Interpolating EOFs (DINEOF) (Hernandez-Carrasco et al., 2018), and Self-
Organizing Maps (SOM) (Hernandez-Carrasco et al., 2018).
Given the motivation described above, and developed partially within the
framework of JERICO-NEXT project, we present a Lagrangian-based Short-Term
Prediction (L-STP from now on) methodology using existing HFR datasets, to be
applied to surface current real-time observations. The proposed L-STP
methodology aims to be capable to use the previously developed gap filling OMA
method and generate forecasts in near-real time with low computational costs,
compared to the previously presented forecast methods, but with the same level of
assessment. The uniqueness of this approach is two-fold: first, the historical
Eulerian velocity fields are used to construct a catalogue of Lagrangian trajectories
and second, using the trajectories obtained from present observations, analogues in
the past dataset are searched in order to obtain the best predictive match. The
method is based on Lagrangian computations which have proven to be robust
against errors in velocity field data and against the dynamics of unresolved scales,
since the averaging effect produced by integrating over trajectories which extend in
time and space, tends to cancel random-like errors (Hernandez-Carrasco et al.,
2011, Sayol et al., 2014). Consequently, they are reliable for the assessment of the
dynamical flow structures.
Analogues is a widely used method in time series prediction, especially in early
weather forecasting and statistical downscaling. It is based on the assumption that
if the behavior of a dynamical system at a given time is similar or close enough to
some other situation in the historical record, then the evolution in the future of the
state of the system will be similar to the evolution observed in the same historical
record. Simply stated, two analogue fields are two distinct fields that are close
enough considering a given metric, to be considered as equivalent. Finding of the
best (nearest) analogue of a specific time does not require a historically continuous
dataset, as long as the dataset contains subsets of observations that extend longer
than the testing period and are representative of the range of potential states that
the system can have. These statistically analogue events occur naturally in the
environment and this methodology has been applied and tested in atmospheric
forecasts (Lorenz, 1969, Jianping et al,1993, Prince and Goswami 2007, Shao and
Li 2013).
It must be stressed that this is the first time that the analogues technique has been
applied to the HFR-derived ocean surface currents to obtain short-term forecast, to
the knowledge of the authors. The L-STP is intended to be implemented
operationally with low computational cost (seconds to few minutes for each
forecast, depending on the size of the historical dataset) and is easily implemented
using existing HFR data processing tools.
**2.   DATA AND METHODS**
*2.1 Data*
HFR data from two distinct oceanographic regions have been used for the
evaluation, validation, and testing of the developed methodology (Figure 1): The
Bay of Biscay (hereinafter BoB HFR) and the central Red Sea region (hereinafter
Red Sea HFR). The range and the spatial resolution of the HFR current systems
depend on their working frequency and the conductivity of the water over which
the system is measuring. Ranges vary from 15 to 220 km range and spatial
resolution from 250 m to 12 km. Typically, a 12 MHz radar has a range ~70 km
with a spatial resolution of 2-5 km. HFR systems usually average current
measurements for one hour, although some average currents for shorter periods,
such as 30 minutes. HFR data from these two regions are used to evaluate the skill
of the method under different dynamical conditions, and with a sufficient set of
observations to provide a database suited to the efficient research of appropriate
analogues. The BoB HFR system, located in the southeastern corner of the Bay of
Biscay, in the Basque Country, is composed of two CODAR Seasonde sites,
working since 2009 at 4.5 MHz frequency, covering up to 200 km range and
providing hourly surface velocity field at 5 km of spatial resolution. The dataset
used in this study spans the period from January 2012 to December 2015. The Red
Sea HFR system is located on the central western coast of Saudi Arabia and is also
composed of two CODAR Seasonde sites. The Red Sea sites are operational since
June 2017, transmit at 16.12MHz frequency, covering up to 120 km range and
providing the hourly surface velocity field at 3 km spatial resolution. The dataset
used in this study spans the period from June 2017 to October 2018.
The BoB HFR has been chosen as the pilot system for testing the developed
methodology, since it has the longest data series and because several papers have
already provided an extensive description of the local circulation and dynamical
processes (Rubio et al., 2013a, 2013b, 2018, 2019, 2020; Solabarrieta et al 2014,
Solabarrieta et al., 2015, Hernandez-Carrasco et al. 2018, Manso-Narvarte et al.,
2018; Declerk et al., 2019). The resulting methodology is then applied to the
operational Red Sea HFR dataset, as a study case. Coastal dynamics in the BoB
show a clear seasonality where cyclonic and anticyclonic eddies dominate in
winter and summer, respectively in responding to local winds and the mean coastal
current (Iberian Poleward Current) (Esnaola et al., 2013, Solabarrieta et al., 2014).
The circulation in the central Red Sea also demonstrates a clear seasonality
(Sofianos and Johns, 2003; Yao et al., 2014a, 2014b; Zarokanellos et al., 2017a,
2017b) linked to the seasonal winds of the area (Abualnaja et al., 2015; Langodan
et al., 2017b). The region is dominated by eddy activity, with both cyclonic and
anticyclonic eddies occurring in the region (Zhan et al., 2014; Zarokanellos et al.
2017a). Due to the only recently available dataset (since mid-June 2017 to present)
the detailed small-scale surface circulation processes of this area is under
characterization at the moment.
The primary difference between the two HFR systems is the operating frequency
resulting in a larger spatial coverage for the BoB HFR than for the Red Sea HFR
and a higher spatial resolution for the latter (5km and 3 km, respectively). This
difference in the spatial resolution should result in better capturing the small-scale

dynamical features in the Red Sea, which could influence the selection of an analogue.

The data from both systems have been processed similarly. The spectra of the received backscattered signal are converted into radial velocities using the MUltiple SIgnal Classification (MUSIC) algorithm (Schmidt 1986). HFR Progs MATLAB package (https://github.com/rowg/hfrprogs) is then used to combine radial currents and generate gap-filled total 2D currents, by means of the Open Modal Analysis (OMA) methodology of Kaplan and Lekien (2007).

*2.2 Lagrangian analogues*

The proposed prediction system, based on the analogue identification method, has been developed with the objective of providing HFR velocity fields forecast (up to 48 hours). As an innovative element, we use a Lagrangian approach in searching for analogues through an historical library composed of particle trajectories, instead of the commonly used Eulerian velocity fields. In our methodology we find the best analogue by comparing maps of trajectories obtained from the last available 48 hours (target field) with the historical catalogue of maps of Lagrangian trajectories (hereinafter Lagrangian catalogue). Then the catalogue map with the trajectory pattern closest to the target field map is selected. Relying on the similar evolution of the current situation and the past analogue, the next 48-hour time velocity fields of the selected analogue provides the target period forecast. In other words, if we find a state in the historical database that is "close enough" to the target field, we assume that the forecast for the current observations will evolve in the same way as did for the chosen analogue. A detailed description of the short-term prediction system is provided in the following algorithm:

1. *Lagrangian catalogue configuration.* First, to build the Lagrangian catalogue, a set of synthetic trajectories was computed by advecting N particles uniformly initialized on a regular grid (Figure 2) in the OMA HFR velocity fields. The N Lagrangian particles are released every hour over the whole available velocity data and are advected during 48 hours. The maps of trajectories of the catalogue are referred as to $\mathbf{X}_C$.
2. *Target map.* A map of trajectories corresponding to the most recent HF currents observations, and referred as to $\mathbf{X}_T$, is computed using the same procedure than for the Lagrangian catalogue but now advecting the N particles in the available last 48 hours ($t_f$–48h) of HFR velocity fields, where $t_f$ corresponds to the current time.

3. *Searching for the analogue.* A searching algorithm for the best (closest to the target map) analogue among all the trajectory maps is implemented next. To increase the efficiency of this process, the search was done in two steps.

    i.  Optimization of the catalogue. First, selecting only "potential" analogues with a similar main drift reduces the Lagrangian catalogue. The trajectories centroid for each map of the catalogue is computed and compared to that of the target field, and finally discarding the analogues whose centroid was at a distance greater than $\delta_{cg}$. The value of the $\delta_{cg}$ is selected to be small enough to minimize the computational time but sufficiently large to do not lose sampling variability in the potential analogues. We explored different values of this threshold distance to find that $\delta_{cg} = 2\xi = 10$ km (where $\xi$ is the spatial resolution) makes a good compromise between computational cost and number of potential analogues in both study areas.

    ii. In a second step, we computed the Lagrangian errors ($\varepsilon$) between the trajectories of the target field and the potential analogues, defined as:

$$\varepsilon_{ANL} = \sqrt{\frac{1}{T}\sum_{j=1}^{T}(\delta_{ANL}(t_i))^2}, \quad t_i = \{6, 12, 24, 36, 48 \text{ hours}\}, \qquad (1)$$

   where $T = 5$ is the number of elements of the set of times $t_i$, and $\delta_{ANL}(t_i)$ is the mean separation distance at time $t_i$ between the trajectories belonging to the target field $\mathbf{X}_T$ and each of the potential analogues $\mathbf{X}_c$, given by:

$$\delta_{ANL}(t_i) = \frac{1}{N}\sum_{j=1}^{N}\left|\left(X_T^j(t_i) - X_c^j(t_i)\right)\right|, \qquad (2)$$

   being N the total number of trajectories j.

4. *Best analogue.* The selection of the best analogue is performed by the Equation (2), which is a simple measure of similarity between two datasets. The best analogue is selected as the element of the catalogue with the lowest $\varepsilon_{ANL}$. Figure 3 shows an example of the time series of $\varepsilon_{ANL}$ values, through the catalogue of potential analogues for a specific case. Then we locate the time $t_{ANL}$ corresponding to best analogue: $t_{ANL} \rightarrow \min(\varepsilon_{ANL}) = \varepsilon_{ANL}(t_{ANL})$ : $X_c(t_{ANL})$.

5. *Currents Prediction.* Once we have identified $t_{ANL}$, the short term forecast of the HFR velocity fields is given by the hourly velocity fields corresponding to the next 48 hours since $t_{ANL}$ (hereinafter "L-STP fields"):
   $\mathbf{X}_{STP}(t_c+1:t_c+48h) = \mathbf{X}_c(t_{ANL}+1:t_{ANL}+48h) \rightarrow \mathbf{V}_{STP}(t_f+1:t_f+48h) = \mathbf{V}_c(t_{ANL}+1:t_{ANL}+48h),$

where $\mathbf{V}_C$ ($t_{ANL}$) is the velocity field corresponding to the best analogue and
$\mathbf{V}_{STP}$ are the forecast currents.

Figure 2 provides an example of the selected analogue (Figure 2b) and
corresponding L-STP fields (Figure 2d) for a given target field (Figure 2a) and the
'truth' trajectories for the following 48 hours from the date of the target field
(Figure 2c). The associated temporal series of errors for the target field and the
potential analogues are shown in Figure 3, where the value of $\varepsilon_{ANL}$ is marked using
a red dot (corresponding to the error between the trajectories of the L-STP field in
Figure 2d and the truth trajectories for the forecast period in –Figure 2c).
To assess the performance of the methodology, we computed forecasted
trajectories based on persistence of currents (hereinafter 'persistence fields' $\mathbf{X}_{PRS}$).
To obtain simulated trajectories using persistence currents, the particles are
advected during 48 hours using a constant (frozen) velocity field (given by the
current velocity field, or target field, $\mathbf{V}(t_f)$) during the 48 hours of simulation:
$\mathbf{V}(x,y,t_f+T) = \mathbf{V}(x,y,t_f)$, where $t_f$ = current time and T={1 : 48h}.
The mean drift of the truth forecasted trajectories, $\mathbf{X}_{TRU}$, is also computed for each
simulation period (the mean drift is computed averaging over all the particle
trajectory length during 48 hours).
The Lagrangian errors between the truth trajectories $\mathbf{X}_{TRU}$ and the L-STP
trajectories $\mathbf{X}_{STP}$ were also computed as:
$$\varepsilon_{STP} = \sqrt{\frac{1}{T}\sum_{j=1}^{T}\left(\delta_{STP}(t_i)\right)^2} = \sqrt{\frac{1}{T}\sum_{j=1}^{T}\left(\frac{1}{N}\sum_{j=1}^{N}\left(X_{TRU}^j(t_i) - X_{STP}^j(t_i)\right)\right)^2}, \qquad (3)$$
where $\delta_{STP}$ is the mean separation distance between truth and the L-STP
trajectories for t= t : t+48 (following 48 hours from the study time). To compare
with persistence, we also compute the Lagrangian error between the truth
trajectories $\mathbf{X}_{TRU}$ and the trajectories derived from the persistence field $X_{PRS}$,
$$\varepsilon_{PRS} = \sqrt{\frac{1}{T}\sum_{j=1}^{T}\left(\delta_{PRS}(t_i)\right)^2} = \sqrt{\frac{1}{T}\sum_{j=1}^{T}\left(\frac{1}{N}\sum_{j=1}^{N}\left(X_{TRU}^j(t_i) - X_{PRS}^j(t_i)\right)\right)^2}, , \qquad (4)$$
where $\delta_{PRS}$ is the mean separation distance between truth maps of trajectories,
$X_{TRU}$, and maps of trajectories from persistent velocity fields, $\mathbf{X}_{PRS}$, for t= t:t+48
(following 48 hours from the study time)

All the process for the selection and validation of the analogue with the different variables has been summarized in Figure 4. The time series and spatial distribution of the $\varepsilon_{STP}$ and $\varepsilon_{PRS}$ errors have been analyzed for both study areas. Finally, $\varepsilon_{STP}$ and $\varepsilon_{PRS}$ time series have also been calculated and compared to the time series of the $\varepsilon_{ANL}$, in order to evaluate if the $\varepsilon_{ANL}$ can be used as an indicator of the expected skill of the L-STP with respect to the persistence.

Some parameters in the algorithm have to be tuned in order to optimize the results and the computational cost. For instance, we found that the optimal number of particle trajectories, N is equal to 25. All the trajectories have been computed considering infinitesimal and passive particles without adding a diffusion term. To this end we used the Lagrangian module included in the HFR_Progs MATLAB package.

The ability of this method relies on the precision in finding two matching HFR currents states over the entire region, which is dependant on the historical record of observations used to build the catalogue and the dynamical representativity of the catalogue. In this study we use four-year dataset (2012-2015) of trajectory maps computed for the SE BoB, where the trajectory maps from the three first years (2012-2014) were used as Lagrangian catalogue, and the remaining year (2015) was used as a test period. The historical Lagrangian catalogue for this HFR system is, thus, composed of 26304 maps of N=25 trajectories of 48-hours. Then the method was applied to the Red Sea dataset, for the period of July 2017-October 2018. As the dataset temporal extension was short (1 year and 4 months), we have used the whole period to build the Lagrangian catalogue and act as a test period at the same time. In this case, for the analogues search the 5-days period around the date of the target field was removed from the catalogue at each iteration, to avoid temporal overlapping with the target field.

## 3. RESULTS

Figure 2 shows an example of the developed methodology applied to the BoB HFR system on April 15, 2015. It is a visual representation of the (a) target trajectories, (b) the selected analogue, (c) truth trajectories during the next 48 hours from the target period, and (d) the L-STP trajectories provided by the method (48 hours from the analogue).

The performance assessment results are described in section 3.1 and the temporal and spatial forecast for both study areas are shown in section 3.2.

Figure 5 shows the $\varepsilon_{ANL}$ through 2015 for the BOB study area, together with the
$\varepsilon_{STP}$ and $\varepsilon_{PRS}$. The analysis of this plot aims to check the relation between $\varepsilon_{ANL}$, $\varepsilon_{STP}$
and $\varepsilon_{PRS}$. Black dots over the timeline in Figure 5 show the times when $\varepsilon_{STP}$ is
higher than the $\varepsilon_{PRS}$, which occurs 12% of the time. The mean value of the $\varepsilon_{PRS}$ is
73% higher than the $\varepsilon_{STP}$. The correlation between $\varepsilon_{ANL}$ and $\varepsilon_{STP}$ is 0.46 while
correlation between $\varepsilon_{ANL}$ and $\varepsilon_{PRS}$ is 0.05, for the whole test year (2015). Focusing
on the times when the $\varepsilon_{PRS}$ is lower than the $\varepsilon_{STP}$, it can be seen that they mostly
occur during winter months. Previous works in this area have shown that there are
high persistent eastward currents that can last for several weeks during winter
months (Solabarrieta et al., 2014), which can explain the better performance of the
persistence fields in this period.
The hourly values of $\varepsilon_{STP}$ and $\varepsilon_{PRS}$ have been plotted against their corresponding
hourly $\varepsilon_{ANL}$ values for the test year, ordered from minimum to maximum along the
x-axis in Figure 6. We observe that, when $\varepsilon_{ANL}$ is low (less than 13.06 km for this
data set), $\varepsilon_{STP}$ is smaller than $\varepsilon_{PRS}$. However, as $\varepsilon_{ANL}$ increases, $\varepsilon_{STP}$ and $\varepsilon_{PRS}$
converge until an inflection point beyond which $\varepsilon_{STP}$ is slightly greater than $\varepsilon_{PRS}$.
For the SE BoB experiment, the inflection point occurs at $\varepsilon_{ANL}$ =13.06 km and 88%
of cumulative $\varepsilon_{ANL}$. Results from the Red Sea HFR system indicates a similar
pattern (not shown), when the inflection point occurs at $\varepsilon_{ANL}$ = 12.81 km and at
86.4% of cumulative $\varepsilon_{ANL}$.
Further analysis to elucidate the mean separation distances ($\delta_{STP}$ and $\delta_{STP}$) related
to $\varepsilon_{ANL}$ after 6, 12, 24, 36 and 48 hours are presented hereinafter. $\varepsilon_{ANL}$ has been
plotted together with the mean separation distances of the trajectories ($\delta_{STP}$ and
$\delta_{PRS}$), after 6, 12, 24, 36 and 48 hours for each target field (Figure 7). $\delta_{STP}$ is always
higher than the $\delta_{PRS}$ for the 6 hours' simulation. But the values of $\delta_{STP}$ show lower
values than $\delta_{PRS}$ for the lowest $\varepsilon_{ANL}$ for the simulations at 12, 24, 36 and 48 hours.
The values of the correlation coefficient ($R^2$) between the $\varepsilon_{ANL}$ and $\delta_{STP}$ and
between $\varepsilon_{ANL}$ and $\delta_{PRS}$ after 6, 12, 24, 36 and 48 hours are summarized in Table 2,
in order to analyze the relations between the Analogue, the L-STP and the
persistence. Values of $R^2$ for $\varepsilon_{ANL}$ and $\delta_{PRS}$ are small (almost no correlation),
varying between 0.01 and 0.11, while correlations between $\varepsilon_{ANL}$ and $\delta_{STP}$ are
higher, varying between 0.19 and 0.56, and showing higher correlation (>than
0.37) after 12 hours of simulations. The behavior of the Red Sea HFR system
figures (not shown) is similar to the BoB HFR system.
Figures 6 and 7 (and the same ones for the Red Sea system, not shown) show that
while $\varepsilon_{ANL}$ increases, $\varepsilon_{STP}$ and $\delta_{STP}$ increase, but $\varepsilon_{PRS}$ and $\delta_{PRS}$ decrease, showing an
inflexion point (hereinafter $\varepsilon_{ANL(*)}$). The $\varepsilon_{ANL(*)}$ can be calculated just for the
historical dataset but $\varepsilon_{ANL}$ can also be calculated in real time and compared with
$\varepsilon_{ANL(*)}$. It gives a reference value for the forecast skills:
$\varepsilon_{ANL} < \varepsilon_{ANL(*)} \rightarrow \quad \delta_{\_STP} < \delta_{\_PRS} \qquad \rightarrow$ Use L-STP
$\varepsilon_{ANL} > \varepsilon_{ANL(*)} \rightarrow \quad \delta_{\_STP} > \delta_{\_PRS} \qquad \rightarrow$ Use Persistence
To assess the capabilities of the L-STP methodology, times when $\varepsilon_{ANL} < \varepsilon_{ANL(*)}$
have been just analyzed from now on, as when $\varepsilon_{ANL} > \varepsilon_{ANL(*)}$ we recommend to
use persistent currents as a short term forecast.

*3.2 Spatio-temporal performances of the L-STP methodology*

Mean separation distances between truth and forecasted trajectories after different
periods of integration times have been computed for both systems just for $\varepsilon_{ANL} <$
$\varepsilon_{ANL(*)}$ times (Figure 6), in order to evaluate the temporal forecast capabilities of the
methodology. Separation distances computed for the whole test year 2015, are
shown in Figure 8, for the BoB HFR observations.
The separation distances between the measured trajectories and predicted persistent
and STP trajectories, have similar values during the first 6 hours (4km) of the
forecast period, with slightly better results for persistent trajectories. But after 6
hours, the separation distance for the forecast based on persistent currents increases
faster than using L-STP. At 24 hours, the separation distance is 11 km for
persistence forecasts and 8km for L-STP forecasts. The values are 12 and 18km,
respectively, after 48 hours of simulation. The mean drift values of the truth
trajectories show that the mean drift is similar to the L-STP separation distances,
during the 48 hours.
Temporal mean separation distances between truth and forecasted trajectories for
the Central Red Sea HFR System, computed for $\varepsilon_{ANL} < \varepsilon_{ANL(*)}$ are shown in Figure
9. The separation distances for the STP forecasts are higher than those forecasts
with persistent currents during the first 15 hours. After 15 hours, quality of
forecasts reversed where STP produced better results than persistence.
Spatial distribution of the difference between $\delta_{PRS}$ and $\delta_{STP}$ at 6, 12, 24 and 48
hours, for the BoB and the Red Sea study areas, are shown in Figure 10 and Figure
11.

For the BoB HFR system, the differences are not appreciated during the first 6
hours. However, after 12 hours of simulation, the advantage of the L-STP is clear
in most of the study area, especially outside the continental shelf slope where
persistent currents dominate the circulation. The separation values between $\delta_{PRS}$
and $\delta_{STP}$ increase up to 10km after 48hours of simulation.
For the Red Sea, the significant differences between STP and Persistence start after
24 hours of simulation, and continue until 48 hours.

**4. DISCUSSION**
In this work, a new methodology to forecast ocean surface currents based on HFR
observations has been described. The approach is based on the search of analogues
in a trajectory (Lagrangian) space using a previously generated trajectory field
catalogue. The temporal and spatial skills of the proposed L-STP methodology
have been analyzed in the previous section.
The target Lagrangian trajectory maps have been compared with the previously
generated trajectory catalogue to obtain $\varepsilon_{ANL}$, $\varepsilon_{STP}$, $\varepsilon_{PRS}$, $\delta_{STP}$ and $\delta_{PRS}$ for each
analyzed time. For the BoB system (2015 period), the correlation between $\varepsilon_{ANL}$ and
$\varepsilon_{PRS}$ is 0.05, showing no relation between them and similar values are obtained for
$\varepsilon_{ANL}$ and $\delta_{PRS}$ (0.01-0.11- from table 2). The correlation between $\varepsilon_{ANL}$ and $\varepsilon_{STP}$ is
0.46 and it varies from 0.19 to 0.56 between $\varepsilon_{ANL}$ and $\delta_{STP}$. Although the
correlation between $\varepsilon_{ANL}$ (past) and $\delta_{STP}$ or $\varepsilon_{STP}$ (future) are low, they suggest that
there is a relation between the errors of the analogues and the errors of the L-STP.
$\delta_{STP}$ is always higher than the $\delta_{PRS}$ for the 6 hours' simulation. Which means that
for the first hour, it is better to use persistence.
The $\varepsilon_{ANL(*)}$ can just be calculated for the historical dataset but $\varepsilon_{ANL}$ can also be
calculated and compared to the previously selected $\varepsilon_{ANL(*)}$, in real time. It gives a
reference value for the forecast skills and we suggest that $\varepsilon_{ANL}$ can be considered as
a real-time skill-score metric for the L-STP :
$$\varepsilon_{ANL} < \varepsilon_{ANL(*)} \rightarrow \quad \delta_{\_STP} < \delta_{\_PRS} \qquad \rightarrow \quad \text{Use L-STP}$$
$$\varepsilon_{ANL} > \varepsilon_{ANL(*)} \rightarrow \quad \delta_{\_STP} > \delta_{\_PRS} \qquad \rightarrow \quad \text{Use Persistence}$$
The election of the best value for $\varepsilon_{ANL(*)}$ is the main sensitive step of the proposed
methodology: the values of $\varepsilon_{ANL}$ are different for each study area and no fixed
value can be given. Due to this, an exhaustive analysis of $\varepsilon_{ANL}$, $\delta_{STP}$, $\delta_{PRS}$ of the
historical dataset is required to find the correct inflexion point and select a correct
$\varepsilon_{ANL(*)}$, before the method can be applied to a new study area.
Once fixed $\varepsilon_{ANL(*)}$, the skills of the proposed L-STP methodology have been tested
in figures 8 to 11. The values of the $\delta_{STP}$, compared to previous works in the BoB
area showed that the L-STP produces accurate predictions, which demonstrates the
ability of the Lagrangian approach to capture key dynamical features needed to
accurately predict the proper dynamical conditions.
For the BoB HFR System, temporal $\delta_{STP}$ shows values of 3.5km, 5.5km and 8km,
after 6, 12, and 24 hours respectively. The $\delta_{STP}$ values are similar to the $\delta_{PRS}$ values
during the first 6 hours of simulation but $\delta_{STP}$ are lower after that, with 3km and
5.5km of difference between them, after 24 and 48 hours of simulation,
respectively (*Figure* 8). As stated in previous work, that the circulation over the
BoB area is dominated by a stable, persistent current field during winter
(Solabarrieta et al., 2014) which is reflected by these results where persistence has
good or even slightly better forecasting skill during the first 6 forecast hours than
the proposed methodology.
$\delta_{STP}$ values for the BoB HFR system are similar to the ones obtained by
Solabarrieta *et al.*, 2016, for the whole year but $\delta_{STP}$ are better for summer months,
for the same study area. They used the linear autoregressive model, described in
Frolov et al., 2012, to forecast HFR current fields and the errors using that
approach were 2.9 and 7.9 km after 6 and 24 hours. Although the results obtained
in this work improve only during certain periods the forecast presented in
Solabarrieta et al., 2016, the presented methodology has three advantages over the
previous method: it is easy to run in real time; it does not require a continuous
training period; and it is able to discriminate the times when the usage of the
persistence is applicable. On the negative side, it requires the generation of a
catalogue of past trajectories as the search space for analogues, but once it is ready,
it is easily increasable in real time, without extra pre-analysis; just adding new
trajectory fields to the previous catalogue.
The values of the $\delta_{STP}$ for the Red Sea HFR system follow a similar pattern to the
BoB results, with higher separation distances. This may be related to the limited
time span of the available dataset, as a better closest analogue may be found in a
longer dataset.

The spatial comparison of the $\delta_{STP}$ and $\delta_{PRS}$ for the BoB HFR system (Figure 10), shows that the L-STP has better skills for the entire study area after 12 hours of simulations. The skills of the L-STP with respect to the persistence increases with time, showing up to 10km of improvement relative to persistence at 48 hours in some parts of the study area. For the spatial distribution, after 12 hours, the smallest differences between $\delta_{STP}$ and $\delta_{PRS}$ occurred over the slope. This is explained by the existence of persistent seasonal Iberian Poleward Current that flows along the continental slope toward the east along the Spanish coast and northward along the French coast (Solabarrieta *et al.,* 2014). In other words: although the L-STP can be performant in periods of persistent currents, the persistence field can show a better forecast for a short temporal scale (48h). L-STP will improve those forecasts, as soon as spatio-temporal variability increases.

The results for the Red Sea HFR system are similar but the benefit of the L-STP methodology appears only after 12 hours of simulation. Spatially, the improvement is again lower where persistent currents occur, as it is the case of the Eastern Boundary Current that flows northward following the eastern Red Sea Coastline in the study area (Bower and Farrah, 2015; Sofianos and Johns, 2003; Zarokanellos *et al.,* 2017b). The dominance of the persistent currents is evident in the lower values of the difference between the STP forecasts and the Persistence forecasts as shown in Figure 11 and in comparison, with Figure 10.

We have compared the capabilities of the L-STP methodology against the forecast based on the persistence of currents. The L-STP method requires long (but not continuous) training periods and improves the results obtained from previously developed HFR forecast system (Solabarrieta et al., 2016) in the same study area (BoB) for the whole year. However, the L-STP still shows some limitations in predicting some specific dynamical scenarios, i.e. the dynamical conditions originated by the persistent IPC (Iberian Poleward Current). We have found that the Lagrangian analogue is not able to properly identify such persistence, it performs relatively better during non-persistent periods. The fact that persistent events in both study areas are characterized by narrow high-speed jets (i.e. IPC in the BoB) small spatial differences in the location of the main circulation could generate high separation distances between the reference and predicted trajectories. While the trajectory computed from the velocity field predicted from the persistence model is advected in the same jet, the currents obtained from the L-STP are slightly shifted, but just enough to advect the particle in a different position within the jet, originating, therefore larger errors (larger εSTP). We have observed that the longer the training period (as in the BoB system), the better the performance of the L-STP method. This suggests that longer training periods

would increasing the capability to identify periods of persistent dynamics occurring over the same area, and thus improving the performance of the L-STP.

As mentioned, previous efforts to forecast surface currents from HFR data have shown similar results compared with the methodology presented in this paper. However, the advantage of the L-STP method is that it can be used in near-real time, with short and non-continuous datasets of around 2-3 years.

## 5. CONCLUSION

A methodology forecast surface currents with analogues of Lagrangian dynamics in real-time has been proposed. This methodology provides accurate forecast of sea surface currents up to 48 hours and its capability has been tested in terms of spatial and temporal distributions. The methodology has been successfully applied to two distinct coastal regions to evaluate its capabilities in different hydrodynamic regimes, although further analysis using data from more areas is required to generalize the methodology.

Relationships between $\varepsilon_{ANL}$ and $\varepsilon_{STP}/$ $\varepsilon_{PRS}$ suggest that the $\varepsilon_{ANL}$ can be considered as a reliable indicator of the method's performance. Taking in consideration all the analyses done in this work, we propose to use STP currents for trajectory or velocity field predictions from 12 hours forward, if the $\varepsilon_{ANL}$ value is lower than $\varepsilon_{ANL(*)}$. If $\varepsilon_{ANL}$ is higher than $\varepsilon_{ANL(*)}$, or the forecast is just for the next 6 hours, the use of the persistence field is suggested. We also suggest that the $\varepsilon_{ANL(*)}$ value and forecast transition time need to be carefully evaluated for each study region. This, of course, infers that a minimum data set is required before the L-STP method can be applied.

Further analysis of analogue finding approaches is required to improve the observed results, especially during periods when currents are persistent. The use of longer dataset as a training period may improve this aspect. Then, the next step would be to test the methodology for additional periods and other regions, to analyze the possibility to find analogues for different sub-regions and to evaluate its functionality in an operational mode.

The methods to find the minimum training period for each system should be analyzed deeper in future works. The minimum training period will be directly related to the variability of the local dynamics and those should be considered during the analysis.

The HFR Progs MATLAB package (https://
cencalarchive.org/~cocmpmb/COCMPwiki) has been used to generate total
currents from radial files and to fill the spatial gaps of the surface current field
using the OMA method, and to generate Lagrangian trajectories. The presented
forecasting method can be therefore easily implemented as an additional tool to
provide short term forecast at the same time that they generate total currents.

**DATA AVAILABILITY**

The Red Sea HF Radar data can be requested through:

- https://lthdatalib.kaust.edu.sa

Historical and NRT Bay of Biscay HF Radar data can be requested through:

- Euskoos portal: https://www.euskoos.eus/en/data/basque-ocean-meteorological-network/high-frequency-coastal-radars/
- Emodnet Physics -

  http://www.emodnetphysics.eu/Map/platinfo/piradar.aspx?platformid=10273

- CMEMS Instac - http://marine.copernicus.eu/services-portfolio/access-to-products/?option=com_csw&view=details&product_id=INSITU_GLO_UV_NRT_OBSERVATIONS_013_048

## AUTHOR CONTRIBUTION

- **Lohitzune Solabarrieta**: She has worked on the set up of the methodology, data analysis, manuscript writing and final submission.

- **Ismael Hernandez-Carrasco**: He has worked on the set up of the methodology and the manuscript writing.

- **Anna Rubio**: She has worked on the set up of the methodology, data analysis, and manuscript writing.

- **Michael Campbell**: He has worked on the configuration of the methodology. He has also contributed on the manuscript writing.

- **Ganix Esnaola**: He has worked on the configuration of the methodology. He has also contributed on the manuscript writing.

- **Julien Mader**: He has contributed on the writing of the manuscript.

- **Burton H. Jones**: He has contributed on the writing of the manuscript.

- **Alejandro Orfila**: He has worked on the configuration of the methodology, data analysis and the manuscript writing.

## COMPETING INTERESTS

The authors declare that we have no conflict of interest

## ACKNOWLEDGEMENTS

This work was funded by a Saudi Aramco-KAUST Center for Marine Environmental Observation (SAKMEO) Postdoc fellowship to Lohitzune Solabarrieta, and from the Integrated Ocean Processes (IOP) Group in KAUST. We acknowledge the support of the LIFE-LEMA project (LIFE15 ENV/ES/000252), the European Union's Horizon 2020 research and innovation program under grant agreement No. 654410 & 871153 (JERICO-NEXT and JERICO-S3 Projects), the Directorate of Emergency Attention and Meteorology of the Basque Government, the MINECO/FEDER Project MOCCA (256RTI2018-093941-B-C31). and the Department of Environment, Regional Planning, Agriculture and Fisheries of the Basque Government (Marco Program). This work was partially performed while A. Orfila was a visiting scientist at the Earth, Environmental and Planetary Sciences Department at Brown University through a Ministerio de Ciencia, Innovación y Universidades fellowship (PRX18/00218). Ismael Hernandez-Carrasco acknowledges the Vicenç Mut contract funded by the Balearic Island Govern and the European Social Fund (ESF) Operational Programme. The HF radar-processing toolbox HFR_Progs use to produce OMA was provided by D. Kaplan and M. Cook, Naval Postgraduate School, Monterey, CA, USA.

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

    **TABLES**

    *Table 1: Characteristics of the previously developed STP works based on HFR data.*

| Authors | Approach | Needs continuous training period | Comple-mentary data required? | Region of application | Reliable forecast period |
|---|---|---|---|---|---|
| Zelenke 2005 | EOF + bilinear regression model | Yes | Wind | Oregon coast | 48 hours |
| Frolov et al. 2012 | EOF + linear auto regression model | Yes | Wind and tides (optional) | Monterey Bay, California | 48 hours |
| Barrick et al., 2012 | Constant linear trend model applied to OMA modes | Yes | Wind | Finnmark, Norway | 12 hours |
| Orfila et al. 2015 | EOF+Genetic Algorithm | Yes | No | Toulon, France | 48 hours |
| Solabarrieta et al. 2016 | Frolov et al., 2012 | Yes | No | Bay of Biscay | 48 hours |
| Vilibić et al., 2016 | SOM+neural network +winds | Yes | Wind | Northern Adriatic Sea | 72 h |
| Ren et al., 2019 | Random Forest (RF) classification algorithm | No | Tide and Wind | Galway Bay, Ireland | 59 h |
| This paper: L-STP | Analogue finding | No | No | Bay of Biscay and the Central Red Sea | 48 h |

*Table 2: Correlation coefficient values between best $\varepsilon_{ANL}$ and $\delta_{\_STP}$ and between $\varepsilon_{ANL}$ and*
*$\delta_{\_PRS}$, after 6, 12, 24, 36 and 48 hours of simulation.*

|  | 6 hours | 12 hours | 24 hours | 36 hours | 48 hours |
|---|---|---|---|---|---|
| $R^2$ $\varepsilon_{ANL} - \delta_{\_STP}$ | 0.19 | 0.37 | 0.55 | 0.56 | 0.54 |
| $R^2$ $\varepsilon_{ANL} - \delta_{\_PRS}$ | 0.07 | 0.11 | 0.03 | 0.01 | 0.04 |
| $\varepsilon_{ANL}$ [km], for the inflection point between $\delta_{\_STP}$ and $\delta_{\_PRS}$ | - | 11.94 | 12.44 | 13.09 | 14.33 |
| % of $\varepsilon_{ANL}$ (accumulative) for the previous line | - | 81 | 84 | 87 | 95 |


  **FIGURES**

*Figure* 1*: (Up) A global view of both analyzed study areas. (Down-Left) HFR*
*system of the BoB. (Down-Right) HFR system of the central Red Sea. Blue dots*
*represent the data points and the black cross are the HFR antenna positions*

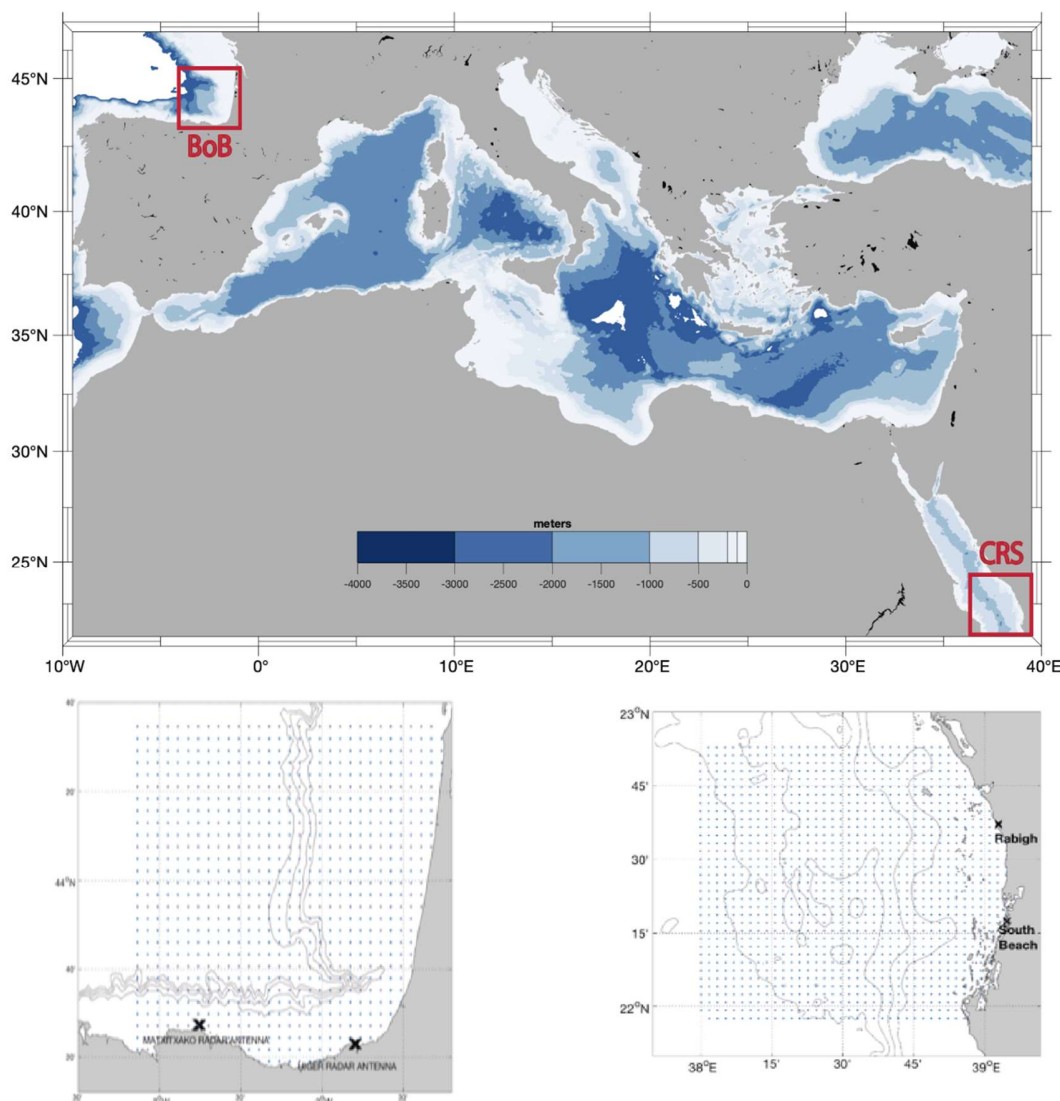


*Figure 2: (1) 15-Apr-2015 00:00 example of the developed methodology applied to*
*the BoB HFR system. (a) The past 48 hours of target field of test period (b) The*
*analogue having the lowest error, (c) The truth trajectories for the forecast period*
*(d) the STP trajectories. The initial positions of the particle trajectories are*
*indicated by the blue dots, and the red dots indicate the position after 48 hours.*

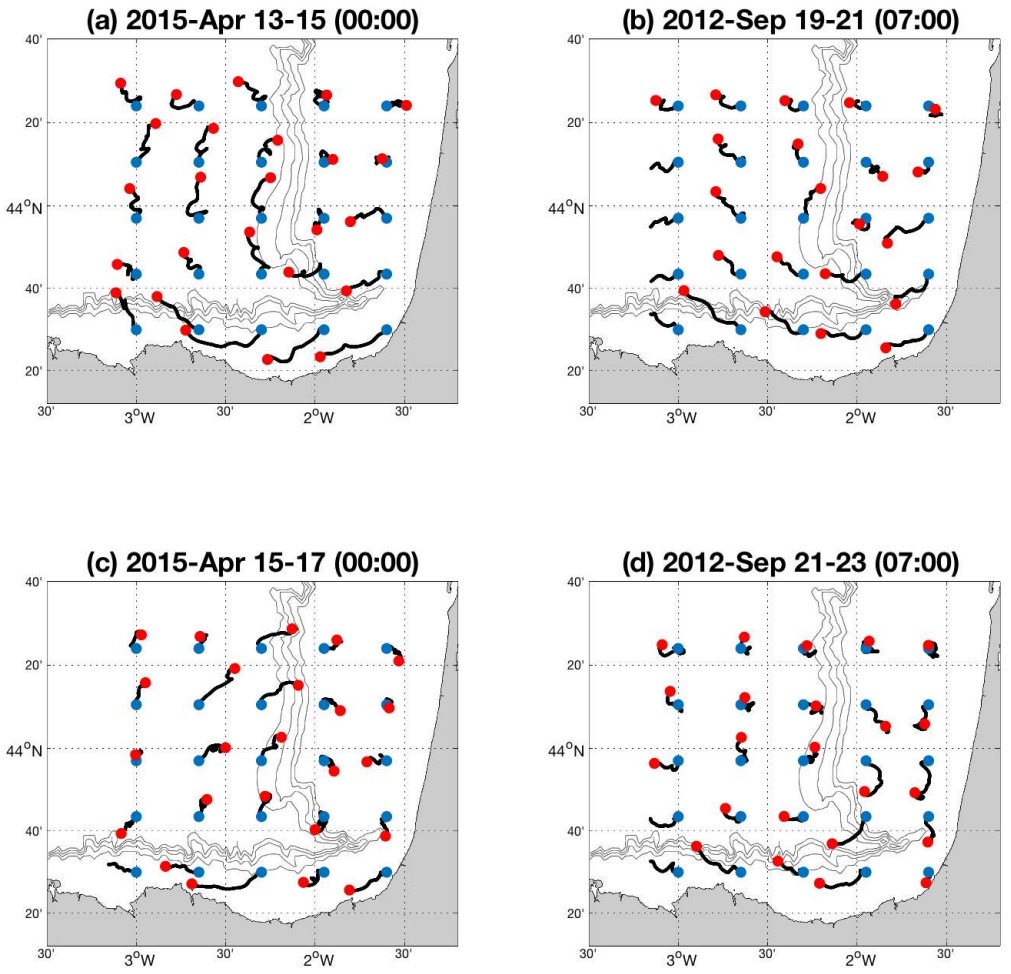


*Figure 3: Example for the test period on 15-Apr-2015 00:00; errors for the whole Lagrangian catalogue fields of the BoB HFR System (training period 2012-2014), restricted to the δ_cg = 10 km condition. The red dot indicates the occurrence date and the error of the best analogue (19-Sep-2012 07:00).*

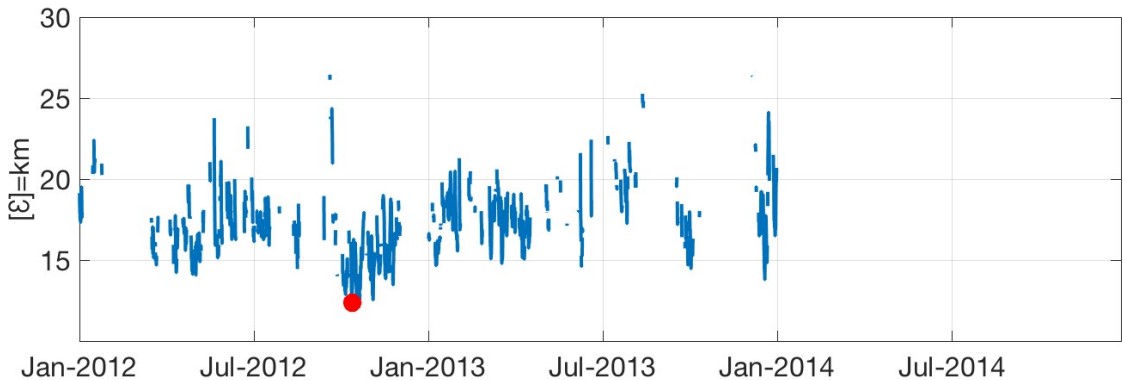

*Figure* 4*: Scheme of the analogue selection and L-STP forecast assessment*
*process.*

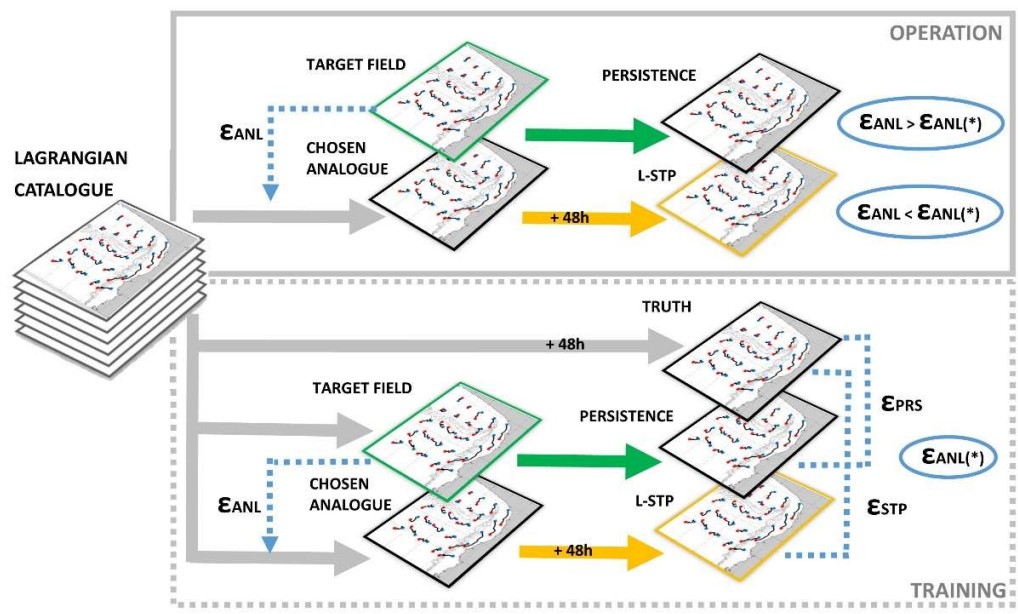



*Figure 5: errors of the hourly best analogue for the BoB HFR, for 2015 ($\varepsilon_{ANL}$),*
*together with the $\varepsilon_{STP}$ and $\varepsilon_{PRS}$. The black dots over the timeline show the times*
*when $\varepsilon_{STP}$ is higher than $\varepsilon_{PRS}$*

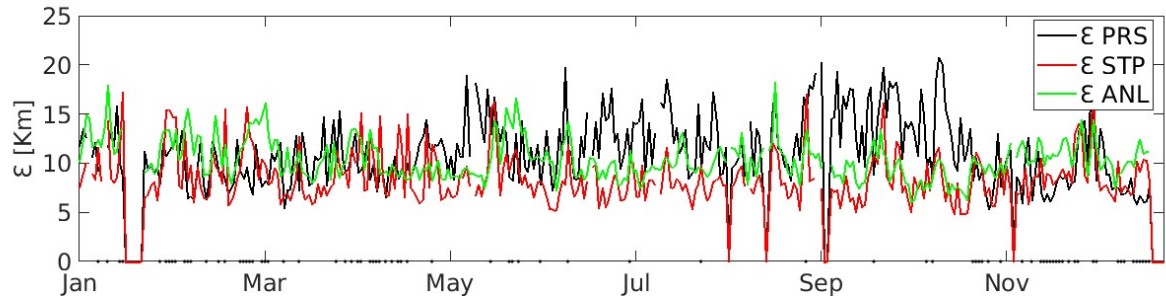


*Figure 6: X axis shows the $\varepsilon_{ANL}$, ordered from minimum to maximum, for the best*
*analogue for the test year 2015, for the BoB HFR. Left Y axis indicates $\varepsilon_{STP}$ (red)*
*and $\varepsilon_{PRS}$ (blue) for the corresponding $\varepsilon_{ANL}$. Right Y axis indicates the % of the*
*accumulative comparison times as shown by the gray solid line. Dashed vertical*
*line indicates the crossing point between $\varepsilon_{STP}$ and $\varepsilon_{PRS}$ ($\varepsilon_{ANL*}$=13.06 km).*

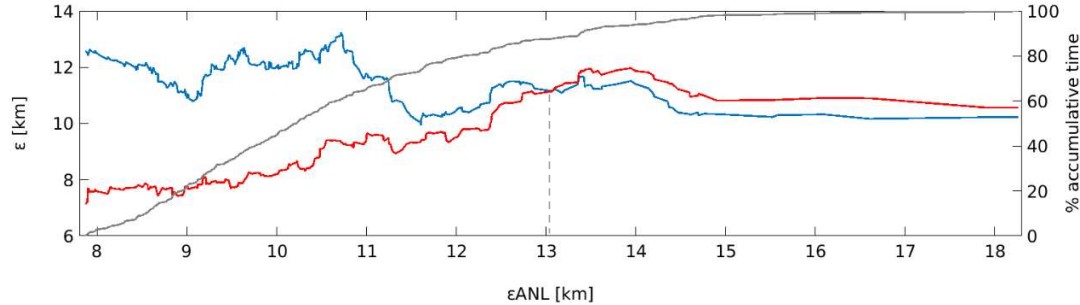


*Figure 7: Left Y axis indicates δ_STP (red) and δ_PRS (blue) for the corresponding*
*ε_ANL, after 6, 12, 24, 36 and 48 hours. Right Y axis is the cumulative % of timesteps*
*in the computation of the mean errors, as indicated by the black line in the plots. X*
*axis is the ε_ANL, ordered from minimum to maximum, for the best analogue for the*
*test year 2015 (BoB HFR system)*

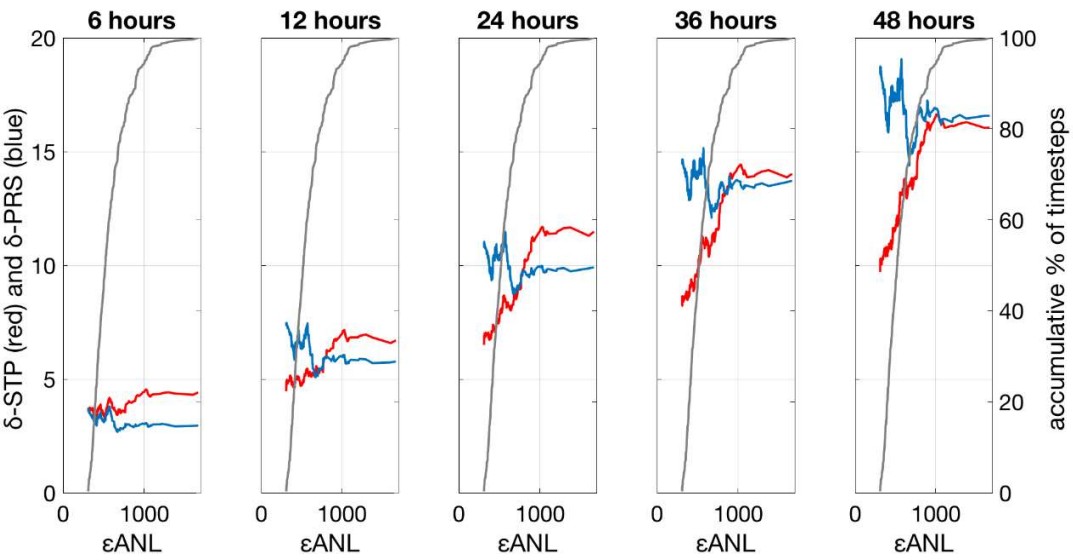


*Figure* 8*: Time evolution of the mean separation δ_STP and δ_PRS [km] between*
*truth and forecast trajectories using truth and STP/PRS currents and the mean*
*drift, with BoB system data, for 2015. The mean drift of the truth forecasted*
*trajectories is also computed for each simulation period (the means drift is*
*considered as the average of the distances moved by each particle during 48*
*hours).*

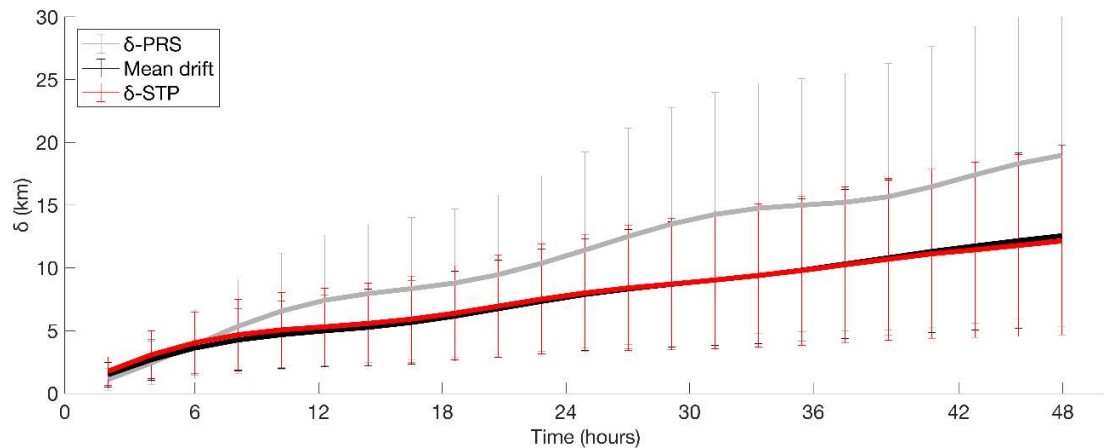


*Figure* 9*: Time evolution of the mean separation distances $\delta_{STP}$ and $\delta_{PRS}$ [km]*
*between real and forecast trajectories using truth and STP/PRS currents and the*
*mean drift, with the Red Sea HFR system data, for July 2017 to October 2018. The*
*mean drift of the truth forecasted trajectories is also computed for each simulation*
*period (the means drift is considered as the average of the distances moved by*
*each particle during 48 hours).*

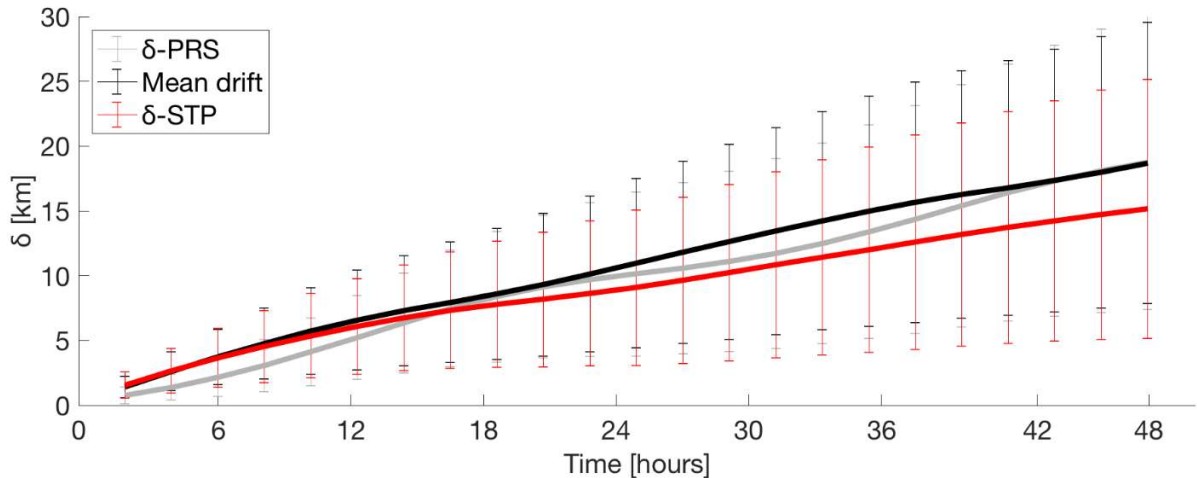


*Figure* 10*: Spatial distribution of separation distances [km] between trajectories*
*using L-STP and persistent currents at 6, 12, 24 and 48 hours, for the BoB HFR*
*System.*

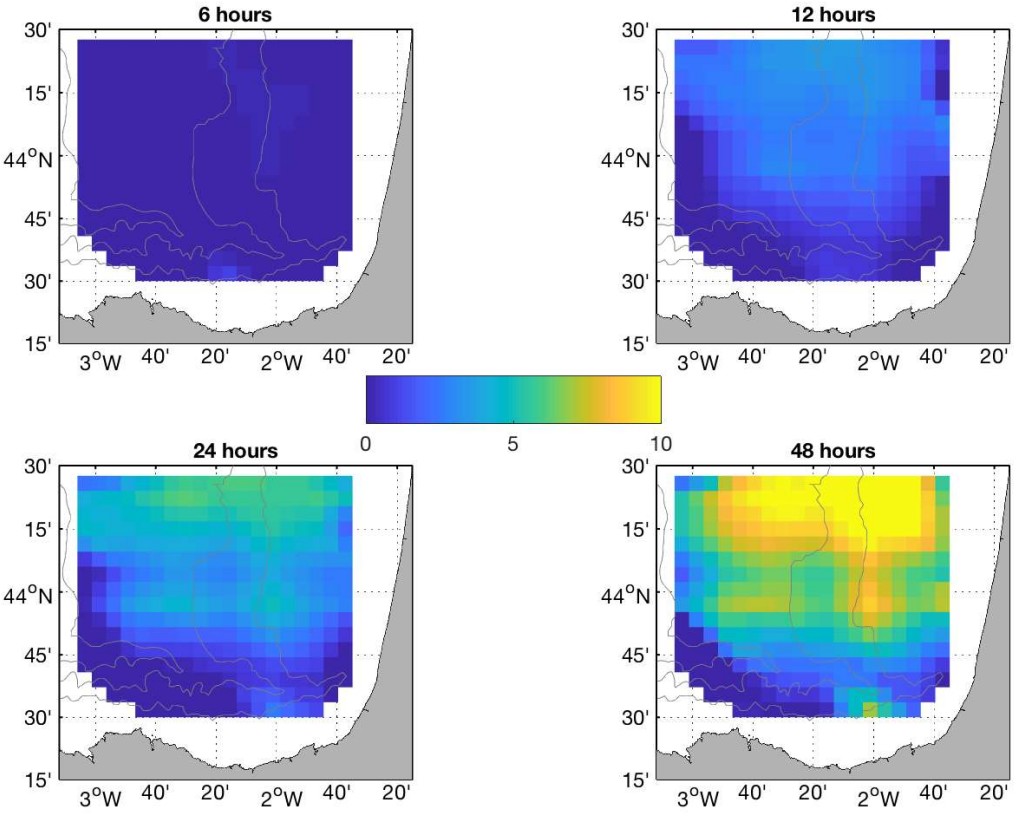


*Figure* 11*: Spatial distribution of separation distances [km] between trajectories*
*using L-STP and persistent currents at 6, 12, 24 and 48 hours, for the Red Sea*
*HFR system.*

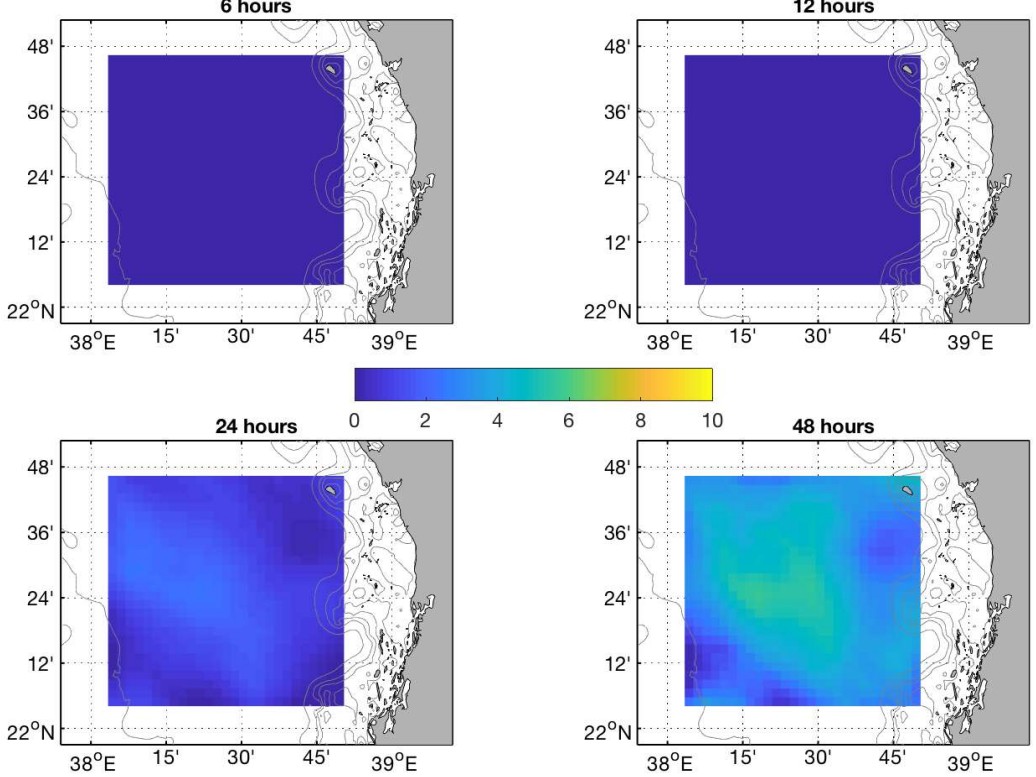
