# Peer review of "A NEW LAGRANGIAN BASED SHORT TERM PREDICTION METHODOLOGY FOR HF RADAR CURRENTS"

_Ocean Science, 2019_

## Referee Comment (RC1) · Anonymous Referee #1 · 12 Mar 2020

The manuscript describes the application of the method of analogues to the prediction of Lagrangian trajectories computed from HFR.

Lagrangian trajectories are computed from an historical data set providing surface currents from HFR systems. The catalogue of these Lagrangian trajectories is the basis to be compared to any new data set, from a present HFR surface currents. Then the future time evolution of the analogue provides the forecast for the present case.

The best analogue is selected in 2 steps. First the difference between the centroid of the 25 trajectories (the 48-h or the end position, is not clear) of each hour of the catalogue is compared with the centroid of the target field. Only the analogues resulting in a difference lower than 10km are selected. Then a Lagrangian error ($\epsilon_{ANL}$) is defined as the sum of the mean separation distance between trajectories computed

from the catalogue fields and those computed from the target field, at 4 different times (6, 12, 24, 36 hours of advection). This error is in km$^2$. The field having the lowest error is selected and will provide the analogue forecast.

Why do we need the first step? I suppose that if \delta_cg is bigger than 10km, then the error is high? Is it for computational issues?

To assess the performance of the method, an equivalent Lagrangian error is computed. I'm not sure that the definitions of the errors (\epsilon_STP and \epsilon_PRS) (line 303-304 308-309) are correct. I think that the authors compute the forecast so next 48 hours instead of last 48 hours. Otherwise, I really misunderstood completely the method, which is possible, according to my numerous questions. For example, on Figure 3, I do not understand why the blue dots are the same in a) and c) (or (b) and (d)). The end points of a) shouldn't be the start points of c)? Either (a) is a backward trajectory plot, and (c) a forward plot, or again I'm missing some fundamental explanation.

So let's assume that the authors were mistaken, and that the performance is evaluate by computing the error on the next 48 hours (forecast), by comparing the original field with the analogue forecast. Another forecast is used for comparison, based on a persistent field (constant velocity field for the future). The time series and spatial distribution of the errors have then been analyzed for 2 regions (Bay of Biscay & Black sea)

Figure 4 shows the time series of the errors ANL,STP and PRS . The black dots over the timeline shows the times the STP error is higher than PRS according to the caption, the other way around in the text (line 328)! At this point I was thinking to give up the reading, too many errors, to complicate to decrypt the manuscript. But let's go on. . ... PRS method seems better during winter period, since high persistent structures are present. The correlation between ANL-STP is 0.46 and ANL-PRS is 0.05. How significant are both values? Are the authors happy with the 0.46 value? Does it mean

something for the methodology?

Then the analysis is done by plotting errors (STP, PRS) or separation distances versus error_ANL comparisons are shown and discussed. Here my question is how reliable are the results in terms of the dynamics. The error values are enormous, hundreds of km$^2$, considering the domain size ($\sim$1.5°*1.5° according to Fig1), and the correlation coefficients quite low (maximum of 0.56 according to Table 2). Maybe a visual and qualitative comparison between the eulerian fields (the winner analogue , its forecast vs the target fields) could give an idea of the performance of the method. The values alone are not enough in my sense to validate the methodology.

Maybe this method is worthwhile to be further investigated, but I would recommend to go through a major review, making the method clearer, making a methodological analysis in parallel to a physical explanation. The methodology should also be more detailed. Results should be better presented to be convincing. The analogue method was developed mainly for meteorological dynamics, which have very different time and spatial scales. Moreover, the application of this method to Lagrangian motion which very often exhibits chaotic behavior, even in regular and simple Eulerian flows, is questionable. A sub region may have analogues in one period, and a distant region another period. The authors may consider to work on sub region, and with a higher number of trajectories.

Specific comments:

- Once the Error is defined (eq.1) no need to repeat it (eq.2 & 3), since the difference between the errors is not the equation, but the field used to compute the trajectories and the separation distance.

- Not sure either that the definition of the time interval in line 293 is correct. Maybe the authors wanted to write v(ti)=v(tf), ti=[tf tf+48] ?

- Please find better definitions, and schematize the method. Instead of realized you

may use truth, as for the twin experiments in data assimilation?

- The authors say that the method has been applied to the eulerian field with unsatis-fying results (no improvement compared to other methods). Can the authors suggest some explanations for this?

- How the trajectories are computed is not explained, since the readers may not know the CODAR package. Are they purely advected? Is there any diffusion term?

- What is the physical significance of the error (thousand of kilometers)?

- What is the distance between initial points?
* * *

---

## Referee Comment (RC2) · Anonymous Referee #2 · 1 May 2020

In the paper by Solabarrieta et al. a new short-term prediction method for surface marine transport is presented. The method is based on Lagrangian "analogues" calculated using velocity data from high-frequency coastal radars located in two different regions: the Bay of Biscay and the Red Sea. New-method errors and predictions are compared with those based on persistence. The performance is comparable to other methods reported in previous literature (e.g. Solabarrieta et al, 2016) as mean separation distances are shown to be similar. The new method can be more easily implemented operationally than the others due to its computational cost, which is allegedly low.

A process of major revisions is suggested to address the following concerns:

1) L123: "well demonstrated results". Please explain why OMA was chosen and quantify the OMA skills providing values and the advantages wrt to other methodologies like DINEOF or SOM.

2) L138-146: not clear paragraph here and the concept may be missed. Are the authors trying to justify the choice of a Lagrangian vs Eulerian approach for the analogues? If so, wouldn't be enough to say that Lagrangian trajectories are direct measurements of transport of substances at sea? And also that they are more dependent on resolution as they are more keen on accumulating errors being integrals of the velocity fields?

3) L151: uniqueness and originality of the work. Authors should clearly state whether or not this is the first application of the method of analogues in the ocean.

4) L156: numbers expressing a quantification of the computational costs for the different methods should be provided here. How long does it take to run this new method wrt the one in Solabarrieta et al (2016)? What about wrt other methods?

5) L162-177: how do resolutions in the two regions compare with the Rossby radii? Are spatial resolutions of the HF radars fine enough to capture the marked seasonal variability of the mesoscale features in the whole year for both regions? Please provide number and quantify.

6) L209: a conceptual question that should be addressed. It is my understanding that the OMA method is based on finding the best combination of geometrical modes in a specific region able to maximize the fit with the observations at a specific time. In a way, isn't the combination and gap-filling technique already based on "analogues" modes? Isn't this procedure already creating analogue situations from a dynamical perspective, introducing a bias when epsilon_ANL is calculated? I guess that the other way to pose the same question is: how sensitive are results to the use of OMA? How much do they change if a simple linear interpolation technique is used instead of OMA?

7) L213: clearly say here that the "most similar" concept will be defined later in the paper.

8) L212-218 and L220-226: more concepts are repeated in both paragraphs. Please combine them and shorten accordingly

9) L228-230: where is this shown? I have the impression that a section has been completely cut off from the paper. This is also related to point 23 below

10) L237: is conceptually correct to use the whole period as a test period and a Lagrangian catalogue at the same time for the Red Sea? How do results change if the first year is used as catalogue and the second year as test period?

11) L244: I would suggest swapping Fig.2 and Fig.3 positions as this latter is introduced in the text before.

12) L269: please remove not needed.

13) L326-330 and Fig.4: contradictions and big confusion here. Not easy to understand whether or not black dots show periods when epsilon_STP is either larger or smaller than epsilon_PRS. My guess is that dots are when errors in the predictions are larger than in the persistence. Please double-check and rephrase the whole paragraph

14) L331: what is the time-scale of the persistence of these currents during winter months?

15) L343: indicatES

16) L349-357, Fig.6 and throughout the manuscript: please use the already introduced notation for the mean separation distance like, for example, $\delta^{STP}_{6h}$ ($\delta^{PRS}_{6h}$) and not STP_dist (PRS_dist).

17) L356: not sure what "especially after 12 hours mean"? Maximum values are at 36h. Do the authors want to say that larger values are reached and remain almost constant after 24h? Please rephrase.

18) L357: it should be also mentioned that at t=6h PRS is always better that STP (Fig.6). However we have a problem here: at t=6h $R^2$ for PRS is is lower than for STP

(0.07 vs 0.19). How is this possible?

19) L364: isn't this choice unfair wrt persistence? Shouldn't we consider all of them for a fair comparison?

20) L371: correct, it should be indeed added that persistence during the first hours is actually slightly better

21) L380-381: why does the mean drift follow more the persistence curve in the Red Sea case?

22) L390: the advantage is not clear as this is the difference between the two, does not necessarily mean that one is better than the other. Please modify Figs.9 and 10 as suggested in point 37 below

23) L404-407: what does this mean? Only Lagrangian analogues are shown in the manuscript. Has a section been cut off from the paper? This is also related to point 9 above.

24) L417: contradiction with L327-328

25) L423: "first and only the first". Not really but please quantify as it looks that for BoB is at least during the first 6h and for the Red Sea at least for the first 15h!

26) L429: not sure about this value as it was reported 853 km2 before (e.g. at L342 and L364)

27) L441: Fig.7 not Fig.4, correct?

28) L447-453: these lines belong more to the introduction. They are also qualitative while differences and comparisons between methods should really be quantified.

29) L463-472 and in general for the whole section: discussion is poor. Why aren't HF radars able to capture currents if they are persistent? I would expect radars not to be able to resolve highly-variable small-scale structures, not persistent features! Not

getting (or buying) the idea that something persistent cannot be seen by analogues. A better dynamical insight is needed and expected in the discussion of the results.

30) Fig.1: can we have GDOP maps in the two regions? Can they help discussion? Asking for more reasons: a) obtained ranges look large compared to the radar system positions and distances between them; b) it would be important to visualize in which areas OMA operations are more to be carried out; c) it would be nice to compare/discuss GDOP maps wrt to the error distributions of the new Figs.9 and 10 (see point 37 below)

31) Fig.2: resolution is really terrible, please increase it. Line should be thicker as in Figs. 7 and 8. Why are there gaps in the blue line? Really confused by the fact that caption is reporting Nov 17 2015 instead of April 13-15, 2015 as in Fig.3.

32) Fig.3: why is this time chosen? Is this a good or bad example?

33) Fig.4: resolution is really terrible, please increase it. Lines should be thicker as in Figs. 7 and 8. I would suggest to put them in three different panels as they mostly overlap. Double-check figure and text for black dots meaning.

34) Figs.5 and 6: resolution is really terrible, please increase it. Lines should be thicker as in Figs. 7 and 8.

35) Figs.7 and 8: rearrange x-axis labels to have 6-h intervals ending at 48h.

36) Fig.8 caption: remove (UP)

37) Figs.9 and 10: both figures need improvements to show the errors and not only their differences. Suggestion is to have a total of 12 panels in each region and show for each time three panels, one with $\delta^{STP}$, the second with $\delta^{PRS}$ and the third one with their difference.

38) Figs.9 and 10: put labels indicating times either on top of each panel or in the right bottom corners, on land

---

## Author Comment (AC1) · 30 Jul 2020

**Reviewer #1**

Dear reviewer,

We would like to show our sincere appreciation for your interest and deep analysis of our manuscript, entitled "A new Lagrangian based short term prediction methodology for HF radar currents". We would also like to thank the comments and suggestions you have proposed, they help us realize the paper needed substantial changes to allow more clarity in the presentation of methods and results. The paper has been revised and carefully modified following your advices and comments. They have undoubtedly helped to improve the quality of this manuscript. Our individualized response to your comments can be found below (in blue color).

You can find the new manuscript and the changes that we have done over it, in the final manuscript document that we will upload to the journal (both new and "track changes versions). Line references included in this document, are referred to the "track changes" version and they will be updated if any additional changes are requested by the editor before the final submission of the revised manuscript.
* * *
The manuscript describes the application of the method of analogues to the prediction of Lagrangian trajectories computed from HFR.

Lagrangian trajectories are computed from an historical data set providing surface currents from HFR systems. The catalogue of these Lagrangian trajectories is the basis to be compared to any new data set, from a present HFR surface currents. Then the future time evolution of the analogue provides the forecast for the present case.

The best analogue is selected in 2 steps. First the difference between the centroid of the 25 trajectories (the 48-h or the end position, is not clear) of each hour of the catalogue is compared with the centroid of the target field. Only the analogues resulting in a difference lower than 10km are selected. Then a Lagrangian error ($\epsilon_{ANL}$) is defined as the sum of the mean separation distance between trajectories computed from the catalogue fields and those computed from the target field, at 4 different times (6, 12, 24, 36 hours of advection). This error is in km$^2$. The field having the lowest error is selected and will provide the analogue forecast.

Why do we need the first step? I suppose that if $\delta_{cg}$ is bigger than 10km, then the error is high? Is it for computational issues?

This step decreases the computation time. It is short (seconds to few minutes, depending on the historical dataset) but in this way, it is even shorter. It is explained in the manuscript, in the lines 292-299.

To assess the performance of the method, an equivalent Lagrangian error is computed. I'm not sure that the definitions of the errors (\epsilon_STP and \epsilon_PRS) (line 303-304 308-309) are correct. I think that the authors compute the forecast so next 48 hours instead of last 48 hours. Otherwise, I really misunderstood completely the method, which is possible, according to my numerous questions. For example, on Figure 3, I do not understand why the blue dots are the same in a) and c) (or (b) and (d)). The end points of a) shouldn't be the start points of c)? Either (a) is a backward trajectory plot, and (c) a forward plot, or again I'm missing some fundamental explanation.

You are right. $\varepsilon_{STP}$ and $\varepsilon_{PRS}$ are computed for forecast trajectories to compare them with realized/true trajectories, this was an unfortunate mistake in the captions. Equations' captions have been modified in the text to clarify it and a schema of all the process has also been included in the manuscript (Figure 4) with the same purpose. It is similar to the one that as you can see below, where t is the study time ant t' is the time of the best analogue. We assume that [t : (t+48)] will behave similar to [t' : (t'+48)].

[Figure]

$\varepsilon$ is used to select the winner/best analogue

$\varepsilon$_STP , $\varepsilon$_PRS , $\delta$_STP and $\delta$_PRSare used to validate the methodology and estimate final error or separation distandes between real and forecast trajectories

STP fields are the forecast of the L-STP methodology

Figure 3: (now Figure 2).  The blue dots are the same in all the subplots; those are the points where we initialize our simulations for 48 hours. They need to have the same starting point to be able to make comparisons between them.

So, let's assume that the authors were mistaken, and that the performance is evaluated by computing the error on the next 48 hours (forecast), by comparing the original field with the analogue forecast. Another forecast is used for comparison, based on a persistent field (constant velocity field for the future). The time series and spatial distribution of the errors have then been analyzed for 2 regions (Bay of Biscay & Black sea).

As pointed in our reply for your previous paragraph, your assumption is right and the performance is evaluated computing the error on the next 48 hours, as this will be the case in real time. And it has been analyzed for 2 regions (Bay of Biscay and Red Sea).

Figure 4 shows the time series of the errors ANL, STP and PRS. The black dots over the timeline shows the times the STP error is higher than PRS according to the caption, the other way around in the text (line 328)! At this point I was thinking to give up the reading, too many errors, to complicate to decrypt the manuscript. But let's go on. . .. PRS method seems better during winter period, since high persistent structures are present. The correlation between ANL-STP is 0.46 and ANL-PRS is 0.05. How significant are both values? Are the authors happy with the 0.46 value? Does it mean something for the methodology?

The black dots over the timeline shows the times when $\varepsilon_{STP} > \varepsilon_{PRS}$, as indicated in the caption. It has been corrected in the text (line 420-421) and it is consistent now.

Regarding the correlation values for $\varepsilon_{ANL} - \varepsilon_{STP}$ and for $\varepsilon_{ANL} - \varepsilon_{PRS,}$ as we are comparing the errors of the past with the errors in the future (from the L-STP), we agree that the 0.46 value is low but significant. We point these values in the description of figure 4 (now converted to figure 5) in the manuscript, just to show that although during persistent periods $\varepsilon_{STP}$ is higher that $\varepsilon_{PRS,}$ $\varepsilon_{PRS}$ it is not correlated at all with the $\varepsilon_{ANL}$ , while $\varepsilon_{STP}$ shows bigger correlation, as expected.

Then the analysis is done by plotting errors (STP, PRS) or separation distances versus error_ANL comparisons are shown and discussed. Here my question is how reliable are the results in terms of the dynamics. The error values are enormous, hundreds of km$^2$, considering the domain size (~1.5°*1.5° according to Fig1), and the correlation coefficients quite low (maximum of 0.56 according to Table 2). Maybe a visual and qualitative comparison between the eulerian fields (the winner analogue, its forecast vs the target fields) could give an idea of the performance of the method. The values alone are not enough in my sense to validate the methodology.

As explained in our previous paragraph, the fact that the maximum correlation values between past $\varepsilon_{ANL}$ and future $\varepsilon_{STP}$ or $\varepsilon_{PRS}$ is 0.56 does not mean that methodology is not working; this comparison has been done to check the goodness of our forecast compared with the past $\varepsilon_{ANL}$ values, and to give an advice to the final user to use Persistence or L-STP as forecast.

Figures 8 and 9 (former 7 and 8) have been generated to assess the performance of the methodology. Those separation distances are similar or even better to previously published and validated results.

Maybe this method is worthwhile to be further investigated, but I would recommend to go through a major review, making the method clearer, making a methodological analysis in parallel to a physical explanation. The methodology should also be more detailed. Results should be better presented to be convincing. The analogue method was developed mainly for meteorological dynamics, which have very different time and spatial scales. Moreover, the application of this method to Lagrangian motion which very often exhibits chaotic behavior, even in regular and simple Eulerian flows, is questionable. A sub region may have analogues in one period, and a distant region another period. The authors may consider to work on sub region, and with a higher number of trajectories.

Following your advice, we have corrected the definition of the errors that we had in the first submitted version of the manuscript. We have also added a figure to make a more detailed and clearer description of the methodology.

As it is indicated in the lines 255-267 of the "track changes" manuscript, the analogues methodology was firstly applied to the Eulerian velocity fields but results were clearly worse. We later applied the method to Lagrangian trajectories as they are direct measurements of transport of substances at sea. The obtained results are similar to previously developed STP works based on HFR data (table 1) so the methodology is working fine. The main advantage of it, it is that it is simple, easily applicable in real time with previously existing codes and we can add the trajectories catalogue as we get new currents. This aspect is now better detailed in the manuscript.

The number of trajectories was widely discussed by the coauthors during the tests of the methodology. A higher number of trajectories increased computational time while the improvement of the methodology was not appreciable.

Finally, your doubt about the sub regions was also discussed by the coauthors during the tests. We tried to decompose analogue finding, not only for different periods, but also for different regions. But we discarded this option, as one of the main goals of the methodology is to give a real time and simple forecast, with low computational cost but good results. As we were interested on this and you have also suggested it, we have included this point as a future work, as it is really interesting.

**Specific comments:**

- Once the Error is defined (eq.1) no need to repeat it (eq.2 & 3), since the difference between the errors is not the equation, but the field used to compute the trajectories and the separation distance.

The three errors are different:

$\varepsilon$ (equation 1): it is the error of the target 48 hours field and each 48 hour fields of the catalogue. There is no forecast or prediction here. [min $\varepsilon$ =$\varepsilon_{ANL}$]

$\varepsilon\_{STP}$ (equation 2): it is the error between the real 48 hours after the target 48hours, and the next 48 hours of the winner analogue (min $\varepsilon$ (=$\varepsilon_{ANL}$) from equation 1) [which is considered as our STP forecast].

$\varepsilon\_{PRS}$ (equation 3): it is the error between the real 48 hours after the target 48hours, and the 48 hours trajectory fields using the study hour as persistent currents [which is considered as our PRS fields].

As explained in previous paragraphs in this document and following your indications, the definitions have been improved in the text and a new figure (figure 4) has been also included to make the methodology clear.

- Not sure either that the definition of the time interval in line 293 is correct. Maybe the authors wanted to write v(ti)=v(tf), ti=[tf tf+48] ?

The equation is correct but it has been completed in the text to make it clearer (lines 385-391)

- Please find better definitions, and schematize the method. Instead of realized you may use truth, as for the twin experiments in data assimilation?

The definitions have been improved and the method has been schematized in the new figure 4.

"Realized" has been swapped by "truth" through the whole manuscript.

- The authors say that the method has been applied to the eulerian field with unsatis fying results (no improvement compared to other methods). Can the authors suggest some explanations for this?

Hourly HF Radar surface current fields for both study areas have more than 1000 nodes in their respective footprint areas. And each of those nodes have longitudinal and latitudinal velocity values. Moreover, the variability associated to those hourly fields is really high and we usually have to filter the data to make long time analysis of the surface currents.

In the other hand, Lagrangian trajectories measure the transport of the substances and our final goal is to minimize the separation distances between the truth and simulated trajectories. This fact, together with a lower variability associated to the Lagrangian fields, could be the reason of the better behavior of the analogue methodology with the Lagrangian fields.

- How the trajectories are computed is not explained, since the readers may not know the CODAR

package. Are they purely advected? Is there any diffusion term?

In the Matlab package used in this paper, particles are advected using the HF radar hourly fields and there is no any diffusion term.

It has been included in the text (line 286)

- What is the physical significance of the error (thousand of kilometers)? - What is the distance between initial points?

The physical significance is the sum of the mean square separation kilometers at 6, 12, 24, 36 and 48 hours. It gives and approximation on how big the separation distance is between the truth and simulated trajectories.

The distance between the initial points is different for both systems:

δ_Lat=0.225 and δ_Lon=0.35 for the BoB

δ_Lat=0.1 and δ_Lon 0.15 for the Red Sea

The initial points and the trajectories to be distributed all around the study area is more important than the separation distance of the initial particles.

---

## Author Comment (AC2) · 30 Jul 2020

**Reviewer #2**

Dear reviewer,

We would like to show our sincere appreciation for your interest and deep analysis of our manuscript, entitled "A new Lagrangian based short term prediction methodology for HF radar currents". We would also like to thank the comments and suggestions you have proposed. The paper has been revised and carefully modified following them. They have undoubtedly helped to improve the quality of this manuscript. Our individualized response to your comments can be found below (in blue color).

You can find the new manuscript and the changes that we have done over it, in the final manuscript document that we will upload to the journal (both new and "track changes versions). Line references included in this document, are referred to the "track changes" version and they will be updated if any additional changes are requested by the editor before the final submission of the revised manuscript.
* * *
In the paper by Solabarrieta et al. a new short-term prediction method for surface marine transport is presented. The method is based on Lagrangian "analogues" calculated using velocity data from high-frequency coastal radars located in two different regions: the Bay of Biscay and the Red Sea. New-method errors and predictions are compared with those based on persistence. The performance is comparable to other methods reported in previous literature (e.g. Solabarrieta et al, 2016) as mean separation distances are shown to be similar. The new method can be more easily implemented operationally than the others due to its computational cost, which is allegedly low.

A process of major revisions is suggested to address the following concerns:

1) L123: "well demonstrated results". Please explain why OMA was chosen and quantify the OMA skills providing values and the advantages to other methodologies like DINEOF or SOM.

This paper is focused on the forecast of the surface currents and not on the gap filling techniques. This is why no more quantification values were included in the text. But we have now modified the text, to indicate that one of the main reasons to use OMAs is that it's well-functioning is demonstrated (Kaplan and Lekien, 2007, Hernández-Carrasco et al., 2018) but also because there are available codes in the HFR_progs package, that allow us not only to generate real time gap-filled fields but also to generate trajectories for our analysis (lines 128-130).

2) L138-146: not clear paragraph here and the concept may be missed. Are the authors trying to justify the choice of a Lagrangian vs Eulerian approach for the analogues? If so, wouldn't be enough to say that Lagrangian trajectories are direct measurements of transport of substances at sea? And also that they are more dependent on resolution as they are more keen on accumulating errors being integrals of the velocity fields?

*We agree in this regard with referee. Accordingly, this paragraph has been rewritten in the manuscript (lines 148-153) as follows:*

*Lagrangian computations have proven to be robust in identifying dynamical flow structures and they are direct measurements of transport of substances at sea. Lagrangian diagnostic will capture dynamical features present in the flow that are not readily apparent in pure velocity. At this point we remark that they are more dependent on resolution since they are more keen on accumulating errors being integrals of the velocity fields*

3) L151: uniqueness and originality of the work. Authors should clearly state whether or not this is the first application of the method of analogues in the ocean.

*It has been clarified in the text that apart from the two-fold approach of the presented method, analogue finding to generate Short Term forecast has still not been applied to HF Radar ocean surface velocity fields (lines 181-182)*

4) L156: numbers expressing a quantification of the computational costs for the different methods should be provided here. How long does it take to run this new method wrt the one in Solabarrieta et al (2016)? What about wrt other methods?

*As it has been included in the text, this forecast can be done in seconds or few minutes (depending on the historical dataset size) (lines 184-185).*

*One of the main differences with the rest of the STP methods, is that this new method is not only fast but it can also modify (increase) the historical dataset (catalogue) with the last information as soon as new data are provided, without any requirement to re-analyze the whole catalogue. This clarification has been included later on in the text (lines 635-637)*

5) L162-177: how do resolutions in the two regions compare with the Rossby radii? Are spatial resolutions of the HF radars fine enough to capture the marked seasonal variability of the mesoscale features in the whole year for both regions? Please provide number and quantify.

*The Rossby first radius of deformation in the red Sea is around 30 km (Zhai and Bower, 2013) and between 20 and 50 km in the BoB (~ 3-8 km over the shelf (Charria et al., 2017)). Since the spatial resolutions of both systems are 3 and 5 km respectively they resolve adequately the mesoscale in both regions.*

6) L209: a conceptual question that should be addressed. It is my understanding that the OMA method is based on finding the best combination of geometrical modes in a specific region able to maximize the fit with the observations at a specific time. In a way, isn't the combination and gap-filling technique already based on "analogues" modes? Isn't this procedure already creating analogue situations from a dynamical perspective, introducing a bias when epsilon_ANL is calculated? I guess that the other way to pose the same question is: how sensitive are results to the use of OMA? How much do they change if a simple linear interpolation technique is used instead of OMA?

As pointed by the reviewer, the OMA method finds the best combination of geometrical modes in a specific region to maximize the fitting to the radar surface velocity observations. But it is not "based" in temporal analogues as this fitting is applied independently to each specific hour field, not related to the previous and later fields. Indeed, the OMA method is applied to radial velocities and it can be applied to spatial gaps (due to range fails for example) where linear interpolation technique could not be applied.

7) L213: clearly say here that the "most similar" concept will be defined later in the paper.

Included in the text (lines 245)

8) L212-218 and L220-226: more concepts are repeated in both paragraphs. Please combine them and shorten accordingly

The text has been reorganized and double concepts have been removed (lines 243-253) to make it clearer for the reader.

9) L228-230: where is this shown? I have the impression that a section has been completely cut off from the paper. This is also related to point 23 below

It has been clarified in the text that those results were done during the analysis for this work but that those results are not shown in this paper.

We want to maintain it there, as the reader may think that the direct application of the methodology to the Eulerian fields could be a better approach but we saw that it is not.

10) L237: is conceptually correct to use the whole period as a test period and a Lagrangian catalogue at the same time for the Red Sea? How do results change if the first year is used as catalogue and the second year as test period?

In the Red Sea case, it was indicated in line 236, that the data availability is from July 2017 to October 2018 (2 years). This is just 1 year and 4 months and it has been corrected in the text (Line 264 in the "track control" version).

Ideally, it would be better to use past data as a training period, like the Lagrangian catalogue used for the Bay of Biscay data (because this is the situation that we will have once this method is applied in real time). But taking in account that we know (from previously published works; not HF Radar data) that there is a clear seasonality in the Red Sea study area, and the HF Radar data availability was short, we have used the whole year as a training and test period, but we have removed the previous 2.5 days and the next 2.5 days to avoid the overlapping.

11) L244: I would suggest swapping Fig.2 and Fig.3 positions as this latter is introduced in the text before.

Figures have been swapped and the references corrected accordingly in the text.

12) L269: please remove not needed.

It has been removed and the magnitude of δ_t has been indicated in line 309

13) L326-330 and Fig.4: contradictions and big confusion here. Not easy to understand whether or not black dots show periods when epsilon_STP is either larger or smaller than epsilon_PRS. My guess is that dots are when errors in the predictions are larger than in the persistence. Please double-check and rephrase the whole paragraph

Your guess is correct. Black dots are plotted for the periods when $\varepsilon_{PRS}$ is lower than the $\varepsilon_{STP}$. It has been corrected in the text (line 421) and it is consistent now.

14) L331: what is the time-scale of the persistence of these currents during winter months?

Rubio et al. (2018, 2019) and Solabarrieta et al. (2014) show that currents during winter months show an eastward flow than can least for several weeks during winter and that these currents are higher than eastward flow present during summer season.

It has been completed in the manuscript, in the first paragraph of section 3.1.

15) L343: indicatES

Corrected in the text (line 445)

16) L349-357, Fig.6 and throughout the manuscript: please use the already introduced notation for the mean separation distance like, for example, \delta^STP_6h (\delta^PRS_6h) and not STP_dist (PRS_dist).

$\delta_{STP}$ or $\delta_{STP}$ has been used for the previos $STP_{dist}$ and $PRS_{dist}$.

It has been modified throughout the whole manuscript and the figures.

17) L356: not sure what "especially after 12 hours mean"? Maximum values are at 36h. Do the authors want to say that larger values are reached and remain almost constant after 24h? Please rephrase.

The idea that authors want to show with the combination of figure 6 and table 2 is that there is no correlation between $\varepsilon_{ANL}$ (used to find the analogue in the catalogue) and $PRS_{dist}$ (distance between real and PRS simulated trajectories); while there is higher correlation between $\varepsilon_{ANL}$ and $STP_{dist,}$ specially after 12 hours of simulation (R2($\varepsilon_{ANL}$ vs $STP_{dist}$) increases rapidly after 12 hours, from 0.37 to 0.54) as indicated in table 2)

It has been clarified in the text (lines 460-461)

18) L357: it should be also mentioned that at t=6h PRS is always better that STP (Fig.6). However we have a problem here: at t=6h R^2 for PRS is is lower than for STP

We have mentioned in the text that PRS at 6 hours is always better than STP (lines 454)

Regarding the correlation, there is no any problem. From our understanding, it means that the $\varepsilon_{ANL}$ is correlated with the STP error (bigger $\varepsilon_{ANL \, will}$ have bigger $\varepsilon_{STP}$ or $STP_{dist}$) but it is no correlated with PRS error, even when persistence is better than the STP. The point here is that during the first 6 hours, it is better to use persistency than the STP. But it is worth it to use STP for longer time forecasts (for example, to predict where a possible oil spill could move).

19) L364: isn't this choice unfair wrt persistence? Shouldn't we consider all of them for a fair comparison?

With this comparison, we want to show the capabilities of the methodology for the times when we consider that the STP will be better than the Persistence ($\varepsilon_{ANL} < 853 km^2$, for BoB case). When $\varepsilon_{ANL} > 853 km^2$, we suggest to use persistent currents

$\varepsilon_{ANL}$ can be considered as a real-time skill-score metric for the L-STP. In fact, this value has been investigated and presented to be able to tell to the final user if our forecast is good enough or not.

20) L371: correct, it should be indeed added that persistence during the first hours is actually slightly better

It has been included in the text (line 510)

21) L380-381: why does the mean drift follow more the persistence curve in the Red Sea case?

It is probably related to temporal size of the HF Radar data availability in the Red Sea case. Longer the dataset, better results will be obtained using the presented L-STP method.

22) L390: the advantage is not clear as this is the difference between the two, does not necessarily mean that one is better than the other. Please modify Figs.9 and 10 as suggested in point 37 below

Figures 9 and 10 have been converted to figures 10 and 11, as we have included a new figure.

This point has been replied in point 37 below.

23) L404-407: what does this mean? Only Lagrangian analogues are shown in the manuscript. Has a section been cut off from the paper? This is also related to point 9 above.

As in the point 9 above, it has been again clarified in the text that those results were done during the analysis for this work but that those results are not shown in this paper.

24) L417: contradiction with L327-328

Corrected in the text (line 421)

25) L423: "first and only the first". Not really but please quantify as it looks that for BoB is at least during the first 6h and for the Red Sea at least for the first 15h!

Quantified in the text (line 578-579)

26) L429: not sure about this value as it was reported 853 km2 before (e.g. at L342 and L364)

It was a typo mistake and it has been corrected in the text (line 585)

27) L441: Fig.7 not Fig.4, correct?

Figure 7, correct. It has been corrected in the text (line 597)

28) L447-453: these lines belong more to the introduction. They are also qualitative while differences and comparisons between methods should really be quantified.

They are qualitative but we would prefer to maintain them there, as it is a comparison between both methodologies.

29) L463-472 and in general for the whole section: discussion is poor. Why aren't HF radars able to capture currents if they are persistent? I would expect radars not to be able to resolve highly-variable small-scale structures, not persistent features! Not (0.07 vs 0.19). How is this possible? getting (or buying) the idea that something persistent cannot be seen by analogues. A better dynamical insight is needed and expected in the discussion of the results.

Since temporal resolution of HF-Radars is hourly, they capture well all scales of interest above hours. This includes persistent currents. The comparison in the discussion is made between the STP system based on radars in front of a prediction made with persistence (in an abuse of language since persistence here means that the prediction for the next hour is simply the velocity measured in the last observation).

There is a reason why persistence is better during persistent periods than STP and it is not that STP does not capture persistence. It is mainly because in both cases (BoB and the Red Sea) the persistent periods show high surface velocities and the persistent structures take place in similar longitude and latitude but not exactly the same positions. A small separation distance between real and analogue fields generate high separation distances between real and simulated trajectories. But it does not happen when the real current field is used as persistent current, as it is located exactly in the place where the persistent structure is located in the study time and it will remain there at least during the first few hours.

This paragraph has been rewritted/completed in the manuscript in order to clarify and provide more dynamical insight of the presented results.

30) Fig.1: can we have GDOP maps in the two regions? Can they help discussion? Asking for more reasons: a) obtained ranges look large compared to the radar system positions and distances between them; b) it would be important to visualize in which areas OMA operations are more to be carried out; c) it would be nice to compare/discuss GDOP maps wrt to the error distributions of the new Figs.9 and 10 (see point 37 below)

Figures 9 and 10 have been converted to figures 10 and 11, as we have included a new figure.

This point has been replied in point 37 below.

31) Fig.2: resolution is really terrible, please increase it. Line should be thicker as in Figs. 7 and 8. Why are there gaps in the blue line? Really confused by the fact that caption is reporting Nov 17 2015 instead of April 13-15, 2015 as in Fig.3.

We used different examples during the writing of the manuscript and we finally did not change the date of the caption. But it is corrected now with the correct date: April 15, 2015.

There are gaps in the blue line because the methodology doesn't calculate the errors when the $\delta\_cg$ > 10 km, as indicated in the text and in the caption of this figure.

We have tried to make the line thicker but we loss the details of the times when the error is not calculated because of the $\delta\_cg$ > 10 km condition, as you can see in the next figure:

[Figure]

Regarding the resolution of the figure, we hope that it is just a problem with the revision version of the manuscript. We will submit a high resolution independent file to the journal for the final publication.

32) Fig.3: why is this time chosen? Is this a good or bad example?

This figure has become figure 2, following your advice.

It has been selected as an example of the good functioning of the methodology. There are better and worse examples and we wanted to show something intermediate.

33) Fig.4: resolution is really terrible, please increase it. Lines should be thicker as in Figs. 7 and 8. I would suggest to put them in three different panels as they mostly overlap. Double-check figure and text for black dots meaning.

We have modified the figure increasing the thickness of the lines. We want to maintain the three lines in just one panel to be able to see the comparison of the values. It is too complicated if we

separate it into 3 panels, as you can see in the next plots:

[Figure]

Regarding the resolution, we will proceed in the same way as with figure 2, to submit the figures with high resolution.

34) Figs.5 and 6: resolution is really terrible, please increase it. Lines should be thicker as in Figs. 7 and 8.

Modified as requested.

35) Figs.7 and 8: rearrange x-axis labels to have 6-h intervals ending at 48h.

The figure has been corrected.

36) Fig.8 caption: remove (UP)

It has been corrected in the caption.

37) Figs.9 and 10: both figures need improvements to show the errors and not only their differences. Suggestion is to have a total of 12 panels in each region and show for each time three panels, one with \delta^STP, the second with \delta^PRS and the third one with their difference.

We generated those figures before the submission of the paper and we decided to show just the difference between δ_PRS and δ_STP, as the purpose of this figures is to show the advantage (when exists) of the L-STP methodology vs the usage of persistent fields. But it may help to the reader to have them, so we could include the δ_PRS and δ_STP panels for each study area, as you suggest, as supplementary material for the paper.

As an example, we show here the results of the Bay of Biscay System:

[Figure]

38) Figs.9 and 10: put labels indicating times either on top of each panel or in the right bottom corners, on land

Times have been included on top of each panel.

**REFERENCES:**

Charria G., Theetten S., Vandermeirsch F., Yelekçi Ö., and Audiffren N. Interannual evolution of (sub)mesoscale dynamics in the Bay of Biscay. Ocean Sci., 13, 777–797, https://doi.org/10.5194/os-13-777-2017, 2017

Hernández-Carrasco, I., Solabarrieta, L., Rubio, A., Esnaola, G., Reyes, E., and Orfila, A.: Impact of HF radar current gap-filling methodologies on the Lagrangian assessment of coastal dynamics, Ocean Sci., 14, 827-847, https://doi.org/10.5194/os-14-827-2018, 2018.

Rubio A., Caballero A., Orfila A., Hernández-Carrasco I., Ferrer L., González M., Solabarrieta L., Mader J. Eddy-induced cross-shelf export of high Chl-a coastal waters in the SE Bay of Biscay. Remote Sensing of Environment 205, pp. 290–304, 2018.

Rubio, A., Manso-Narvarte, I., Caballero, A., Corgnati, L., Mantovani, C., Reyes, E., Griffa, A., and Mader, J.: The seasonal intensification of the slope Iberian Poleward Current, in: Copernicus Marine Service Ocean State Report, J. Oper. Oceanogr., Issue 3, 13-18, doi: 10.1080/1755876X.2019.1633075, 2019.

Solabarrieta, L., Rubio, A., Castanedo, S., Medina, R., Charria, G., Hernández, C.: Surface water circulation patterns in the southeastern Bay of Biscay: new evidences from HF radar data. Cont Shelf Res 74:60–76 doi:10.1016/j.csr.2013.11.022, 2014.

Zhai P. and Bower A. The response of the Red Sea to a strong wind jet near the Tokar Gap in summer. Journal of Geophysical Research: Oceans, Vol. 118, 422–434 https://doi.org/10.1029/2012JC008444

---

## Author Response (AR2)

Dear Editor,

We would like to express our sincere appreciation to the reviewers and the Editor for their interest, patience and deep analysis of our manuscript, entitled "A new Lagrangian based short term prediction methodology for HF radar currents". We would also like to thank the comments and suggestions they have proposed. The paper has been revised and carefully modified following those suggestions. They have undoubtedly helped to improve the quality of this manuscript.

Our individualized response to the first revision of Reviewer #1 and the second revision of Reviewer #2 can be found below (the location of the main changes in the text is also indicated).

Hoping the manuscript fulfils now the quality requirements of Ocean Science Journal, I look forward to hearing from you at your earliest convenience.

Yours sincerely,

frue

Lohitzune Solabarrieta

**Reviewer #1**

We thank the effort made by the Referee in reviewing our Manuscript (hereinafter Ms.) entitled *"A new Lagrangian based short term prediction methodology for HF radar currents"*. Taking in consideration the comments of the reviewers, we wanted to make a deep review to present an improved version of the Ms. We have updated the replies for the revision done by the Reviewer #1, accordingly to this new version of the Ms.

We deeply think that thanks to your comments the new version of the Ms. has improved significantly. The paper has been revised and carefully modified following your advices and comments. In the following, you can find the answers to your queries (revision received on 12-Mar-2020) as well as the changes performed in the new version.

**Reviewer #1 Comments & Replies**

The manuscript describes the application of the method of analogues to the prediction of Lagrangian trajectories computed from HFR.

Lagrangian trajectories are computed from an historical data set providing surface currents from HFR systems. The catalogue of these Lagrangian trajectories is the basis to be compared to any new data set, from present HFR surface currents. Then the future time evolution of the analogue provides the forecast for the present case.

The best analogue is selected in 2 steps. First the difference between the centroid of the 25 trajectories (the 48-h or the end position, is not clear) of each hour of the catalogue is compared with the centroid of the target field. Only the analogues resulting in a difference lower than 10km are selected. Then a Lagrangian error (\epsilon\_ANL) is defined as the sum of the mean separation distance between trajectories computed from the catalogue fields and those computed from the target field, at 4 different times (6, 12, 24, 36 hours of advection). This error is in km2. The field having the lowest error is selected and will provide the analogue forecast.

**Comments:**

**Q1. Why do we need the first step? I suppose that if \delta\_cg is bigger than 10km, then the error is high? Is it for computational issues?**

**R1. As the reviewer states this step decreases significantly the computational time. This issue has been explicitly stated in the Ms. (line 212-225).**

"To increase the efficiency of this process, the search was done in two steps..."

**Q2. To assess the performance of the method, an equivalent Lagrangian error is computed. I'm not sure that the definitions of the errors (\epsilon\_STP and \epsilon\_PRS) (line 303-304 308-309) are correct. I think that the authors compute the forecast so next 48 hours instead of last 48 hours. Otherwise, I really misunderstood completely the method, which is possible, according to my numerous questions. For example, on Figure 3, I do not understand why the blue dots are the same in a) and c) (or (b) and (d)). The end points of a) shouldn't be the start points of c)? Either (a) is a backward trajectory plot, and (c) a forward plot, or again I'm missing some fundamental explanation.**

**R2. The referee is right.  $\epsilon_{STP}$  and  $\epsilon_{PRS}$  are computed for forecast trajectories to compare them with realized/true trajectories, this was an unfortunate mistake in the captions. A schema of the process has also been included in the manuscript (Figure 4) to clarify the methodology. Errors have been redefined and the whole section has been rewritten. Errors are now defined in the new Ms (lines 256-277) as:**

"To assess the performance of the methodology, we computed forecasted trajectories based on persistence of currents (hereinafter 'persistence fields'  $X_{PRS}$ ). To obtain simulated trajectories using persistence currents, the particles are advected during 48 hours using a constant (frozen) velocity field (given by the current velocity field, or target field, V(tf)) during the 48 hours of simulation: V(x,y,tf+T)=V(x,y,tf), where tf = current time and T={1 : 48h}.

The mean drift of the truth forecasted trajectories, XTRU, is also computed for each simulation period (the mean drift is computed averaging over all the particle trajectory length during 48 hours).

The Lagrangian errors between the truth trajectories XTRU and the L-STP trajectories XSTP were also computed as:

$$\varepsilon_{STP} = \sqrt{\frac{1}{T} \sum_{j=1}^{T} \left( \delta_{STP}(t_i) \right)^2} = \sqrt{\frac{1}{T} \sum_{j=1}^{T} \left( \frac{1}{N} \sum_{j=1}^{N} \left( X_{TRU}^j(t_i) - X_{STP}^j(t_i) \right) \right)^2}, \quad (3)$$

where  $\delta_{STP}$  is the mean separation distance between truth and the L-STP trajectories for t= t : t+48 (following 48 hours from the study time). To compare with persistence, we also compute the Lagrangian error between the truth trajectories XTRU and the trajectories derived from the persistence field XPRS,

$$\varepsilon_{PRS} = \sqrt{\frac{1}{T} \sum_{j=1}^{T} \left( \delta_{PRS}(t_i) \right)^2} = \sqrt{\frac{1}{T} \sum_{j=1}^{T} \left( \frac{1}{N} \sum_{j=1}^{N} \left( X_{TRU}^j(t_i) - X_{PRS}^j(t_i) \right) \right)^2} , \qquad (4)$$

where  $\delta_{PRS}$  is the mean separation distance between truth maps of trajectories,  $X_{TRU}$ , and maps of trajectories from persistent velocity fields,  $X_{PRS}$ , for t= t:t+48 (following 48 hours from the study time)".

Regarding Figure 3 (now Figure 2), the blue dots are the same in all the subplots; those are the points where we initialize our simulations for 48 hours. They need to have the same starting point to be able to make comparisons between them.

**Q3. So, let's assume that the authors were mistaken, and that the performance is evaluated by computing the error on the next 48 hours (forecast), by comparing the original field with the analogue forecast. Another forecast is used for comparison, based on a persistent field (constant velocity field for the future). The time series and spatial distribution of the errors have then been analyzed for 2 regions (Bay of Biscay & Black sea).**

**R3. As pointed in our reply for your previous paragraph, your assumption is right and the performance is evaluated computing the error on the next 48 hours, as this will be the case in real time. And it has been analyzed for 2 regions (Bay of Biscay and Red Sea).**

**Q4. Figure 4 shows the time series of the errors ANL, STP and PRS. The black dots over the timeline shows the times the STP error is higher than PRS according to the caption, the other way around in the text (line 328)! At this point I was thinking to give up the reading, too many errors, to complicate to decrypt the manuscript. But let's go on. . .. PRS method seems better during winter period, since high persistent structures are present. The correlation between ANL-STP is 0.46 and ANL-PRS is 0.05. How significant are both values? Are the authors happy with the 0.46 value? Does it mean something for the methodology?**

**R4. In Figure 5 (formerly 4), the black crosses over the timeline in the x-axis shows the dates when  $\varepsilon_{\text{STP}} > \varepsilon_{\text{PRS}}$ , as indicated in the caption. It has been corrected in the text (line 314-315).**

"Black dots over the timeline in Figure 5 show the times when  $\varepsilon_{STP}$  is higher than the  $\varepsilon_{PRS}$ , which occurs 12% of the time."

Regarding the correlation values for  $\varepsilon_{ANL} - \varepsilon_{STP}$  and for  $\varepsilon_{ANL} - \varepsilon_{PRS}$ , as we are comparing the errors of the past with the errors in the future (from the L-STP), we agree that the 0.46 value is low but we find it as significant. We point these values in the description of Figure 5 in the manuscript, just to show that even for persistent periods  $\varepsilon_{STP}$  is higher than  $\varepsilon_{PRS}$ ,  $\varepsilon_{PRS}$  it is not correlated at all with the  $\varepsilon_{ANL}$ , while  $\varepsilon_{STP}$  shows bigger correlation, as expected.

**Q5. Then the analysis is done by plotting errors (STP, PRS) or separation distances versus error\_ANL comparisons are shown and discussed. Here my question is how reliable are the results in terms of the dynamics. The error values are enormous, hundreds of km2, considering the domain size ( $\sim 1.5^{\circ} * 1.5^{\circ}$  according to Fig1), and the correlation coefficients quite low (maximum of 0.56 according to Table 2). Maybe a visual and qualitative comparison between the eulerian fields (the winner analogue, its forecast vs the target fields) could give an idea of the performance of the method. The values alone are not enough in my sense to validate the**

methodology.

**R5. Note that errors have been redefined in Section 2.2. As explained in our previous Reply, the fact that the maximum correlation values between past  $\varepsilon_{ANL}$  and future  $\varepsilon_{STP}$  or  $\varepsilon_{PRS}$  is 0.56 does not mean that the method is not providing good forecast. The comparison has been done to compare the forecast against past  $\varepsilon_{ANL}$  values, as well as to provide a warning on the use of Persistence or L-STP as forecast.**

Figures 8 and 9 (former 7 and 8) show the performance of the methodology. The separation distances obtained are similar or even better than previously published and validated results.

**Q6. Maybe this method is worthwhile to be further investigated, but I would recommend to go through a major review, making the method clearer, making a methodological analysis in parallel to a physical explanation. The methodology should also be more detailed. Results should be better presented to be convincing. The analogue method was developed mainly for meteorological dynamics, which have very different time and spatial scales. Moreover, the application of this method to Lagrangian motion which very often exhibits chaotic behavior, even in regular and simple Eulerian flows, is questionable. A sub region may have analogues in one period, and a distant region another period. The authors may consider to work on sub region, and with a higher number of trajectories.**

**R6. We totally agree with the Referee. The Ms. has been fully revisited and specifically and following your advice, the definition of the errors corrected. We have also added a figure to make a more detailed and clearer description of the methodology. We think that one of the advantages of the presented methodology is that it is simple, easily applicable in real time and immediately updated, as new data will become available. All these points have been detailed in the new Ms. The use of a large number of trajectories was widely discussed by the coauthors and tested during the development of the methodology. A higher number of trajectories increased computational time while the improvement of the methodology was not appreciable.**

Regarding the sub regions it was also investigated during the tests. We tried to decompose analogue finding, not only for different periods, but also for different regions. Finally, we discarded this approach, since one of the main goals of the methodology is to provide a fast, reliable real time forecast. However, since we also agree in that this is a very good suggestion, we have included this point as a future work.

Also consider that analogue detection based on full area analogues implies the search in a space that considers all the simultaneous spatial variability, rather than local details, but as the analogue used for forecasting contains small scale and local features, the smallest scale features can be understood and the fingerprint of the bigger scale match.

**Specific comments:**

**Q7. Once the Error is defined (eq.1) no need to repeat it (eq.2 & 3), since the difference between the errors is not the equation, but the field used to compute the trajectories and the separation distance.**

**R7. We totally agree. A completely new section 2.2., unifies the definitions.**

**Q8. Not sure either that the definition of the time interval in line 293 is correct. Maybe the authors wanted to write v(ti)=v(tf), ti=[tf tf+48]?**

**R8. Yes, it has now been modified.**

**Q9. Please find better definitions, and schematize the method. Instead of realized you may use truth, as for the twin experiments in data assimilation?**

**R9. Thank you for your comment (see R#3, #4 and new Figure 4). "Realized" has been replaced by "truth" through the whole manuscript.**

**Q10. The authors say that the method has been applied to the eulerian field with unsatisfying results (no improvement compared to other methods). Can the authors suggest some explanations for this?**

**R10. Hourly HF Radar surface current fields for both study areas have more than 1000 nodes in their respective footprint areas. And each of those nodes has longitudinal and latitudinal velocity values. Moreover, the variability associated with those hourly fields is really high and we usually have to filter the data to make long time analysis of the surface currents. By contrast, Lagrangian trajectories are robust against errors in the velocity field data and against the dynamics of unresolved scales, since the averaging effect is produced by integrating over trajectories which extend in time and space, that tends to cancel random-like errors. Furthermore, we consider that it is better to deal directly with trajectories since our goal is to minimize the separation distances between the truth and simulated trajectories. This fact, together with a lower variability associated with the Lagrangian fields, could be the reason for the better behavior of the analogue methodology with the Lagrangian fields.**

**Q11. How the trajectories are computed is not explained, since the readers may not know the CODAR package. Are they purely advected? Is there any diffusion term?.**

**R11. In the Matlab package used in this paper, particles are considered infinitesimal and neutrally buoyant and are advected using the HF radar hourly fields without any diffusion term. It has been included in the text (line 286).**

**Q12. What is the physical significance of the error (thousand of kilometers)? - What is the distance between initial points?**

**R12. The physical significance is the sum of the mean square separation kilometers at 6, 12, 24, 36 and 48 hours. It gives an approximation on how big the separation distance is between the truth and simulated trajectories. The distance between the initial points is different for both systems:**

 $\delta\_\text{Lat=0.225}$  and  $\delta\_\text{Lon=0.35}$  for the BoB

 $\delta\_Lat=0.1$  and  $\delta\_Lon~0.15$  for the Red Sea

The initial locations and the trajectories to be distributed around the study area is more important than the separation distance of the initial particles.

**Reviewer #2**

We thank the effort made by the Referee in reviewing for second time our Manuscript (hereinafter Ms.) entitled *"A new Lagrangian based short term prediction methodology for HF radar currents.* Taking in consideration all the comments of the reviewer (both in revision 1 and revision 2), we made a deeper review, to present an improved version of the Ms.

We deeply think that thanks to all your comments, this new version of the Ms. has improved significantly. The paper has been revised and carefully modified following your advices and comments. In the following, you can find the answers to your queries as well as the changes performed in the new version.

**Reviewer #2 Comments & Replies**

The new version of the paper by Solabarrieta et al. is overall satisfying but I am still unconvinced by the following concerns:

**Q.a) L250: why are values different from those at lines 221 and 225.**

**R.a: It was a typo error. The frequency is mentioned correctly in the new version of the Ms.**

**Q.b) L282-284: still puzzled by the fact that the authors want to mention the Eulerian attempt. In order to show that the Lagrangian results are better than the Eulerian ones, as argued in their response, authors should clearly explain and provide details on how the analogue for the surface Eulerian velocity is found and at least report quantitative metrics for the comparison.**

**#**R.b: We agree with referee that a clearer explanation was needed. We have chosen the Lagrangian approach because it has been shown that errors in the velocity field and missing inter-grid dynamics effects are reduced in the Lagrangian computations (see Hernandez-Carrasco et al., 2011, Hernandez-Carrasco et al., 2018). Accordingly, this paragraph has been rewritten in the manuscript (lines 112-114) as:

"The method is based on Lagrangian computations since they have proven to be robust against errors in the velocity field data and against the dynamics of unresolved scales, since the averaging effect produced by integrating over trajectories which extend in time and space, tends to cancel random-like errors (Hernandez-Carrasco et al., 2011,Hernandez-Carrasco et al., 2018, Sayol et al., 2014). Consequently, they are robust in identifying dynamical flow structures.

For the sake of clarity, and in order to focus only on the new approach we have removed in the new version of the Ms. all comparison with the Eulerian approaches. This is a very relevant issue that deserves a detailed analysis in a new work.

**Q.c) L330: As the method runs in a very short time it should not be a problem to explore its sensitivity to the number of virtual trajectories used. Looking at Figure 1, we still have plenty of gridpoints. what if we use 100 or 50 or less (like 12) virtual particles? How do results change?**

**R.c: The number of particle trajectories is chosen based on a compromise between the maximum area covered for the trajectories and a reasonable computational cost for operational purposes. We find that the optimal number of particle trajectories, with an affordable operational processing time, was 25. This is indicated in the new Ms., in the lines 290-291. The spatial distribution of the virtual trajectories aims to cover the whole study area. The optimal number of particle trajectories, with an operational processing time was 25, as indicate in the new Ms., in the lines 290-291.**

Using more than 25 virtual trajectories, the improvement in the results was not appreciable but the processing time increased substantially. This is why we found that N=25 was the optimal number.

**Q.d) L348: why is epsilon calculated as the sum of squared difference every 6h? do results change if difference are calculated at every hour?**

**R.d: The results were not differing too much when calculating  $\mathcal{E}$  every hour or every 6h. But the computation time was much shorter, and this is why we decided to use values every 6 hours.**

**Q.e) L574: why is 0.39 meaningful? where is this value?**

**R.e: The correct value is 0.37 (from table 2) and it has been corrected in the new Ms.**

There is no correlation between  $\varepsilon_{ANL}$  (used to find the analogue in the catalogue) and  $\delta_{PRS}$  (distance between real and PRS simulated trajectories); while there is higher correlation between  $\varepsilon_{ANL}$  and  $\delta_{\_STP}$ , especially after 12 hours of simulation (R2 ( $\varepsilon_{ANL}$  vs  $\delta_{\_STP}$ ) increases rapidly after 12 hours, from 0.37 to 0.54) as indicated in table 2. This has been clarified in the text (lines 392-401).

**Q.f) L670: where is shown that the cross point is at 714 km2?**

This value has been removed from the Ms, mainly because the errors have been redefined and a  $\varepsilon_{ANL(*)}$  (directly related with the cross point) has been introduced for a better understanding of the methodology.

**Q.g) Overall discussion: I must admit that this part remains poor. Still not able to grasp why HF radars are less able to resolve persistent feature wrt the PRS method. Why is the PRS method less affected if the persistent structures do no take place at the same positions as the authors are saying in their response? A clear reference to an example where both method are used for one of these structures is really needed. Authors are also not showing GDOP maps which could help the discussion and the interpretation of the results.**

Since temporal resolution of HF-Radars is hourly, they capture well all scales of interest above hours. This includes persistent currents. The comparison in the discussion is made between the STP system based on radars in front of a prediction made with persistence (in an abuse of language since persistence here means that the prediction for the next hour is simply the velocity measured in the last observation). At this point we remark that persistence is defined in the Eulerian frame and not in the Lagragian one which is the one that the present method works. This means that any (small) difference in one of each velocity component could lead to a different trajectory although velocities are persistent. This paragraph has been rewritten/completed in the manuscript in order to clarify and provide more dynamical insight of the presented results.

g) Figure 4: I must confess that I find this figure more confusing than inspiring. It is not really understandable, very wordy and it is not clear what the different lines are referring to.

**R.g: We agree on this comment too. This is why we have generated a new figure to replace the old one and help to the new schematic summary of the methodology, included in 2.2 Lagrangian analogues. We hope that you will find it clear now.**

**Typos:**

1) L73: correct infrastructure

It has been corrected in the new Ms.

2) L76: correct existing

It has been corrected in the new Ms.

3) L148: correct ITS

The sentence has been rewritten in the new Ms.

4) L240: remove one THE

It has been corrected in the new Ms.

5) L331: missing parenthesis

The sentence has been rewritten in the new Ms.

6) L400: here and throughout the manuscript, should not be 'TRUE' and not 'TRUTH' trajectories?

We have preferred to maintain "Truth" throughout the manuscript, as suggested by the reviewer #1 and agreed by the native English speaker coauthors of this Ms.

7) L489 and L494: slightly change notion here as we delta\_t you refer to two different errors

 $\delta_{ANL}$ ,  $\delta_{STP}$  and  $\delta_{PRS}$  have been included to clarify the equations.

8) L530: correct LAST

It has been corrected in the new Ms.

---

## Author Response (AR3)

Revised version, 12 March, 2021

Dear Editor,

We would like to express one more time our sincere appreciation to the reviewers and the Editor for their interest and deep analysis of our manuscript, entitled "A new Lagrangian based short term prediction methodology for HF radar currents". The paper has been revised and carefully modified following their comments and suggestions. They have undoubtedly helped to improve the quality of this manuscript.

Our response to the revision of Reviewer #2 can be found below.

Hoping the manuscript fulfils now the quality requirements of Ocean Science Journal, I look forward to hearing from you at your earliest convenience.

Sincerely,

Lohitzune Solabarrieta

**Reviewer #2**

We thank once again, the effort made by the Referee in reviewing our Manuscript (hereinafter Ms.) entitled *"A new   Lagrangian based short term prediction methodology for HF radar currents"*. The paper has been modified, taking into consideration the last advices and comments raised by the reviewer, in particular, improving the Discussion section.

Please find in the following, the answers to the reviewer queries (comments submitted on 26-Feb-2020), as well as the changes performed in the new version.

**Reviewer #2 Comments & Replies to the revision submitted on 26 Feb 2021**

**Reviewer #2 comments**

I thank the authors for considering my points in the previous rounds of revisions. I acknowledge their efforts in:

a) inserting a map in Figure 1;
b) remaking Figure 4;
c) removing the Eulerian attempt and
d) better explaining the whole developed methodology in the new Section 2.

The manuscript is strongly improved even though I still believe that the discussion part is poor and remain unconvinced by the arguments provided by the authors in the new response, specifically at point #Q.g. I am getting that, in an abuse of language, predictions made with "persistence" use repeatedly and artificially the velocity measured in the last observation. But it is difficult to me to understand how this repeated and artificial velocity can perform better than the velocity measured by radars, since "they capture well all scales of interest above hours". As suggested earlier, both GDOP maps and a clear example showing the accumulation of small differences using the two methods are really needed.

**Typos:**
1) L221: correct implementeD
2) L808: correct globaL
3) L845-848 and L852-855: set font to italic for consistency with the rest of the caption

**Comments & replies from the authors to reviewer #2:**

We agree with the reviewer that the comparison between our prediction system and persistence model may be difficult to interpret. It has also generated several discussions between the coauthors during the analysis and preparation of the manuscripts.

The main idea that we would like to clarify is that the proposed methodology works well, even during the periods when forecast using persistent currents is better. Indeed, the presented methodology is capable to detect those periods when predictions based on persistence perform better than the L-STP and guide the user to use this first solution as the final prediction. We think that this real time capability to detect those periods is a specific skill of the suggested methodology, which improves the final prediction; not the opposite.

Persistence model is referred as when predicting the velocity field at time "t+1" it is simply used the velocity field at time "t" as its prediction. Persistence model is often used as a reference for the assessment of the skill of a prediction system. In our case, we find that the L-STP model performs better than the persistence model, except for a very few and short time periods characterized by persistent dynamical conditions (12%). In these cases, why the persistence model is a better prediction than the "best analogue" of the LSTP model? Why the LSTP model is not able to reproduce (or predict) "persistence" over these time periods when persistent currents control the dynamics?

We would like to clarify this point. The key is that the persistence model prediction is better than the L-STP, during periods of specific persistent dynamics (in both study areas), characterized by intense and shear currents, like the Iberian Poleward Current in the Bay of Biscay area. This is reflected in figure 5 of the Ms. (copied below), where most of the times (black dots over 0 value) occur from November to April, when high persistent currents, linked to the IPC, dominates the dynamics.

[Figure]

*Figure 5: errors of the hourly best analogue for the BoB HFR, for 2015 ($\varepsilon_{ANL}$), together with the $\varepsilon_{STP}$ and $\varepsilon_{PRS}$. The black dots over the timeline show the times when $\varepsilon_{STP}$ is higher than $\varepsilon_{PRS}$*

This dynamical scenario of persistent and intense current periods is recurrent in time, and it mostly occurs in winter. The different events of intense periods share similar characteristics with narrow high-speed jets but slightly different positions, as shown in the next figure:

[Figure]

In this sense, when analogue method finds the most similar Lagrangian field, it will find a similar characteristics pattern period in the Lagranigan catalogue, with the smallest spatial displacement. But those small spatial differences are large enough to generate high errors in the STP method as the separation distances are directed linked to the different speed and direction between the predicted and real currents. However, the repetition of the last hour currents (persistence model prediction) will have lower separation distances as the location of the main jet is more similar.

The predictability, or the growth of a small perturbation in this dynamical system, is given by the growth rate of the separation between two trajectories advected in two different velocity fields (reference and predicted) being higher when the difference between the currents (reference and predicted) are bigger (different direction and speed), that is, as they are more uncorrelated. In our case, under the persistent dynamical conditions governed by the IPC, the currents obtained from the LSTP are slightly shifted, but just enough to advect the particle in a different jet, while the velocity field predicted from persistence advects the particle in the same current.

We have observed that the longer the training periods (as in the BoB system), the better the performance of the STP method. This suggests that longer training periods would improve the capabilities of the STP, increasing the probability to find periods of persistent dynamics which occurred almost in the same spatial area.

We agree with the reviewer that this was not clearly explained in the Ms. and following his/her advices, the next paragraph has been included in the "Discussion" section of the Ms. We think that clarifying the main reasons of the good behavior of the persistence in some specific cases, greatly improves the discussion about the limitations and advantages of the LSTP method.

> *"…*
>
> *We have compared the capabilities of the L-STP methodology against the forecast based on the persistence of currents. The L-STP method requires long (but not continuous) training periods and improves the results obtained from previously developed HFR forecast system (Solabarrieta et al., 2016) in the same study area (BoB) for the whole year. However, the L-STP still shows some limitations in predicting some specific dynamical scenarios, i.e. the dynamical conditions originated by the persistent IPC (Iberian Poleward Current). We have found that the Lagrangian analogue is not able to properly identify such persistence, it performs relatively better during non-persistent periods. The fact that persistent events in both study areas are characterized by narrow high-speed jets (i.e. IPC in the BoB) small spatial differences in the location of the main circulation could generate high separation distances between the reference and predicted trajectories. While the trajectory computed from the velocity field predicted from the persistence model is advected in the same jet, the currents obtained from the L-STP are slightly shifted, but just enough to advect the particle in a different position within the jet, originating, therefore larger errors (larger $\varepsilon_{STP}$). We have observed that the longer the training period (as in the BoB system), the better the performance of the L-STP method. This suggests that longer training periods would increasing the capability to identify periods of persistent dynamics occurring over the same area, and thus improving the performance of the L-STP.*
>
> *As mentioned, previous efforts to forecast surface currents from HFR data have shown similar results compared with the methodology presented in this paper. However, the advantage of the L-STP method is that it can be used in near-real time, with short and non-continuous datasets of around 2-3 years.*
>
> *…"*

We hope that these new paragraphs clarify all the concerns that the reviewer could have in the previous versions about the performance of the methodology.

Apart from this, and following the recommendation of the reviewer, we have calculated the theoretical GDOP maps for the BoB system, as shown in the next figures:

[Figure]

The reader can think that the GDOP could be related to the "worse" performing of the L-STP methodology vs. persistency, during persistent events. But the area where IPC main jet occurs, is a low GDOP error area (as shown in the previous figures), so GDOP error should not add uncertainty to the L-STP methodology.

We think that these figures could generate more confusion on the final reader than clarification, as it is not the reason of the good performing of the persistency. We also think that the included

new explanatory paragraphs at the end of the "Discussion" section, clarify the reasons of the good results of the persistency vs. L-STP during persistent events. Due to all these reasons, we prefer not to include GDOP figures in the Ms, but they could be attached in complementary material, if needed.

**Typos:**

1) L221: correct implementeD

"D" has been introduced in the Ms.

2) L808: correct globaL

It has been corrected in the Ms.

3) L845-848 and L852-855: set font to italic for consistency with the rest of the caption

It has been changed for consistency in the Ms.